# Cis-perturbation of cancer drivers by the HTLV-1/BLV proviruses is an early determinant of leukemogenesis

Nicolas Rosewick[1,2,*], Keith Durkin[1,*], Maria Artesi[1,*], Ambroise Marçais[3], Vincent Hahaut[1], Philip Griebel[4], Natasa Arsic[4], Véronique Avettand-Fenoel[5], Arsène Burny[2], Carole Charlier[1], Olivier Hermine[3,6], Michel Georges[1] & Anne Van den Broeke[1,2]

Human T-cell leukaemia virus type-1 (HTLV-1) and bovine leukaemia virus (BLV) infect T- and B-lymphocytes, respectively, provoking a polyclonal expansion that will evolve into an aggressive monoclonal leukaemia in ~5% of individuals following a protracted latency period. It is generally assumed that early oncogenic changes are largely dependent on virus-encoded products, especially TAX and HBZ, while progression to acute leukaemia/lymphoma involves somatic mutations, yet that both are independent of proviral integration site that has been found to be very variable between tumours. Here, we show that HTLV-1/BLV proviruses are integrated near cancer drivers which they affect either by provirus-dependent transcription termination or as a result of viral antisense RNA-dependent cis-perturbation. The same pattern is observed at polyclonal non-malignant stages, indicating that provirus-dependent host gene perturbation contributes to the initial selection of the multiple clones characterizing the asymptomatic stage, requiring additional alterations in the clone that will evolve into full-blown leukaemia/lymphoma.

[1] Unit of Animal Genomics, GIGA-R, Université de Liège (ULg), Avenue de l'Hôpital 11, B34, Liège 4000, Belgium. [2] Laboratory of Experimental Hematology, Institut Jules Bordet, Université Libre de Bruxelles (ULB), Boulevard de Waterloo 121, Brussels 1000, Belgium. [3] Service d'hématologie, Hôpital Universitaire Necker, Université René Descartes, Assistance publique hôpitaux de Paris, 149-161 rue de Sèvres, Paris 75010, France. [4] Vaccine and Infectious Disease Organization, VIDO-Intervac, University of Saskatchewan, 120 Veterinary Road, Saskatoon, Canada S7N 5E3. [5] Laboratoire de Virologie, AP-HP, Hôpital Necker-Enfants Malades, Université Paris Descartes, Sorbonne Paris Cité, EA7327, 149 rue de Sèvres, Paris 75010, France. [6] INSERM U1163-ERL8254, Institut Imagine, 24 B Boulevard du Montparnasse, Paris 75010, France. * These authors contributed equally to this work. Correspondence and requests for materials should be addressed to A.Vd.B. (email: anne.vandenbroeke@bordet.be).

An estimated 10–20 million people are infected with the Human T-cell leukaemia virus type-1 (HTLV-1) worldwide[1]. In ~5% of infected individuals the virus provokes adult T-cell leukaemia/lymphoma (ATL), an aggressive T-cell malignancy with a poor prognosis[2]. Bovine leukaemia virus (BLV) is closely related to HTLV-1 and causes a very similar B-cell leukaemia in cattle and sheep[3,4]. The virus infects ~50 million dairy cattle worldwide inducing substantial economic costs in infected herds[5]. Like HTLV-1 in humans, following a long period of asymptomatic infection (several years in cattle, several decades in humans) ~5% of BLV-infected animals develop leukaemia/lymphoma. In addition to infecting bovines, it is possible to experimentally infect sheep with BLV, providing a powerful model for studying deltaretrovirus-induced tumours[6,7]. In contrast to cattle, infected sheep systematically develop leukaemia/lymphoma and in a shorter time frame (~20 months). A further advantage of this model is that it is possible to monitor animals from before infection to terminal leukaemia/lymphoma, recapitulating many of the stages observed in HTLV-1-associated human malignancy.

Despite a long history of study in both the HTLV-1 and BLV models, key steps on the road from initial infection to leukaemia development remain poorly elucidated. In chronic stages of infection, HTLV-1 and BLV propagate primarily through clonal expansion of infected T- or B-cells, respectively, resulting in the presence of multiple clones of varying abundance each uniquely identified by their proviral integration site in the host genome. Following a protracted incubation period, one of these clones expands, leading to the accumulation of malignant cells in the peripheral blood (leukaemia) and/or diverse tissues (lymphoma)[4,8–10]. Tumour cells consist of a predominant malignant T- or B-cell clone and chiefly harbour a single integrated provirus, yet integration sites are very variable[10–12]. As a consequence, it has been widely believed that virus-encoded products drive clonal proliferation and influence oncogenic progression. Historically the focus of research in BLV/HTLV-1 has been on the oncogenic potential of the viral TAX protein. TAX can immortalize rodent cells in vitro and induces tumours in transgenic mice, supporting the hypothesis that it is an essential contributor to oncogenesis[13,14]. TAX activates transcription of the provirus and of many host genes, promotes cell-cycle progression and interacts with DNA repair mechanisms[15,16]. However, the lack of TAX (and other viral sense transcript) expression in the majority of BLV/HTLV-1-induced malignancies and the high frequency of proviruses containing alterations inactivating TAX points to a more complicated picture[17–20]. While TAX expression provides a proliferative advantage to the infected clone, it also makes this clone a target for cytotoxic immune response[21]. Therefore, it may be advantageous for the virus to evade the strong immune response to TAX by silencing expression from the positive strand. It is thus widely accepted that TAX fulfils an essential role at early stages of the oncogenic process, yet is not required for late-stage precipitation to monoclonal malignancy.

Over the last years it has become increasingly apparent that another viral product, HTLV-1 basic leucine zipper (bZIP) factor (HBZ) encoded from the minus-strand, plays an important role in the life cycle and oncogenic potential of the virus. HBZ downregulates HTLV-1 transcription, promotes T-cell proliferation and displays oncogenic properties in transgenic mice, suggesting a critical role in HTLV-1-mediated leukemogenesis[16,22]. In contrast to TAX, HBZ is consistently expressed in infected cells in vivo, regardless of their transformation status. HBZ appears to exert its effects through both the transcript and the protein[23]; however, the precise mechanisms by which HBZ contributes to the oncogenic process remain largely unknown. Like HTLV-1, BLV expresses antisense transcripts, AS1 and AS2, driven by 3′LTR-dependent promoter activity and constitutively produced in leukaemic cells[24]. In addition, BLV strongly expresses RNA polymerase III-dependent microRNAs overlapping AS1 and contributing to ~40% of microRNAs in the tumour cell[25].

In the fraction of infected individuals who do progress, many years separate the initial infection from the development of leukaemia/lymphoma. This indicates that infection with BLV/HTLV-1 is not sufficient to provoke tumour development and that secondary events are required to make the transition to a neoplasm. A recent study examined the landscape of mutations in ATLs and found frequent alterations enriched in T-cell related pathways and immunosurveillance[11]. As regards BLV-induced tumours, beyond limited studies that reported frequent genome instability and mutation of p53 (refs 26,27), the occurrence of secondary events in BLV malignancies remains largely unexplored.

As BLV and HTLV-1 vary little in sequence both within and between hosts, and as wide variations exist in clone abundance between infected individuals and over time, it is hypothesized that the proviral integration site is the principal attribute that distinguishes one infected cell clone from the other, thus is a key element on the road to malignancy. Using a quantitative high-throughput sequencing (HTS) approach to characterize the genomic environment of the provirus, previous studies have addressed the role of the genomic integration site in determining clonal expansion and the potential for malignant transformation of cells carrying integrated HTLV-1 or BLV[8–10,28,29]. Although HTLV-1 and BLV preferentially integrate in transcriptionally active genomic regions, near transcriptional start sites and transcription factor-binding sites, there was no reported evidence of recurrent proviral integration. In tumours, there were no clear hotspots of HTLV-1/BLV integration associated with leukaemic clones, although the ontology of the nearest downstream gene was associated with malignant clones in 6% of the ATL cases[10,12].

Despite the variability of integration sites in fully transformed cells, a role for proviral integration and cis-perturbation of host genes in HTLV-1/BLV-induced clonal expansion cannot be excluded. Here, to explore this hypothesis, we carried out RNA-seq of primary tumours in both the human disease and the animal model in combination with HTS mapping of proviral integration sites. We show that HTLV-1/BLV proviruses are integrated in the vicinity of cancer drivers, which they perturb either by provirus-dependent transcription termination or as a result of viral antisense RNA-dependent cis-perturbation. The same pattern is observed at asymptomatic stages of the disease, indicating that provirus-dependent host gene perturbation triggers initial amplification of the corresponding clones, requiring additional alterations to develop full-blown leukaemia/lymphoma.

## Results

We obtained samples from 44 adult T-cell leukaemias/lymphoma (ATLs) (from 35 patients, HTLV-1, human disease) and 47 B-cell leukaemias (from 43 individuals, BLV, animal model). The animal sample set included 15 bovine tumours (natural disease) and 32 tumours from BLV-infected sheep, a well-established experimental model for BLV/HTLV-1 (refs 6,7) (Supplementary Data 1). We utilized stranded RNA-seq data (91 tumours) in combination with an improved version of DNA-seq based high-throughput mapping of integration sites (56 tumours) to simultaneously profile proviral integrations, measure clonal abundance and identify virus–host transcriptional interactions in tumours[9,24,29–31]. We identified

**Table 1 | HTLV-1/BLV interacting host genes are enriched in cancer driver genes.**

| | Gene number | Enrichment (P-values) | | | |
| --- | --- | --- | --- | --- | --- |
| | | Random* | Random Para† | Expr‡ | Expr Para† |
| *Tumour samples*§ | | | | | |
| Genic proviruses | 41 | 0.0029 | 0.0016 | 0.0219 | 0.0111 |
| All proviruses excluding genic concordant poly-(A) | 65 | 0.0004 | <1e-05 | 0.0006 | <1e-05 |
| Intergenic proviruses excluding exon capture | 30 | 0.0015 | 0.0012 | 0.0006 | 0.0004 |
| All proviruses | 82 | <1e-05 | <1e-05 | 0.0004 | <1e-05 |
| | | | | | |
| Asymptomatic samples§ | 723 | <1e-05 | <1e-05 | <1e-05 | 0.0002 |

*Random simulated gene sets.
†Simulated gene sets that include information about paralogs.
‡Expression-matched simulated gene sets.
§Gene subsets as defined in Supplementary Data 3 and Supplementary Table 4.

92 distinct proviral integration sites considering all available tumours (54 genic and 38 intergenic, Supplementary Table 1). Of the examined tumour samples, 87.8% were characterized by a single predominant malignant clone defined by a single integration site and clonal abundance ranging from 79.4 to 99.9%. For the remaining tumour clones ∼6.75% showed evidence of two integrations, while ∼4.05% carried three or four integration sites, with each integration displaying equivalent abundance characteristic of multiple proviruses in a single clone rather than the co-occurrence of multiple tumour clones. Finally, one tumour showed evidence of seven integrations (Supplementary Table 1 and Supplementary Data 2). These observations are consistent with previous findings of multiple proviruses in 9–15% of ATL cases[10,11,32]. The integration sites were defined 86% of the time by both 5′ long-terminal-repeat (LTR)-flanking and 3′LTR-flanking host sequences and 14% of the time by 3′LTR-flanking sequences only (29% of ATLs), in agreement with earlier reports describing 5′LTR-defective (but never 3′LTR-defective) proviruses in both ATLs and B-cell tumours[11,17,32].

**Preferential integration of HTLV-1/BLV near cancer drivers.** A naïve examination of the proviral integration sites revealed a number of striking instances of repeated integration into the same genomic vicinity. Three B-cell leukaemias had BLV integration sites falling within an ∼80 kb region upstream of *FOXR2* (M251, M138 and M21, OAR3.1 chrX: 47,545,987, 47,600,711 and 47,618,373, respectively). Additionally, two pairs of ATLs had integration sites in the same genomic region (ATL1_ly/ATL2_7 within 300 kb of *RBFOX1*, ATL1_ly/ATL58_23 within 600 kb of *CA10*). Finally, a bovine B-cell/ATL tumour pair had integration sites 17 and 13 kb from the same host gene, respectively (T15, ATL4_2, *TMEM67*), while an ovine B-cell/ATL tumour pair had their proviruses integrated within and 430 kb downstream of the same host gene, respectively (M395/ATL25_10, *ELF2*). These observations contrast with previous studies that did show preferential integration in transcriptionally active regions, but failed to observe hotspots of HTLV-1 integration in ATLs[9–11,28]. We showed by simulation that the observed degree of recurrent integration into narrow genomic regions significantly exceeded expectation assuming random proviral integration ($P = 0.0024$; Supplementary Fig. 1). Consistent with this conclusion, one of the ovine B-cell tumours showed evidence of BLV integration into *UBASH3B*, which was identified as the target gene of HTLV-1 integration in one of the ATLs analysed by WGS in the recent study of Kataoka *et al.*[11].

The observed non-random distribution can either indicate that some sites are more prone to proviral integration, possibly reflecting specific chromatin features[28], or/and reflect the fact that integration at specific sites promotes tumorigenesis, i.e., selection. To distinguish between these hypotheses, we verified whether the 54 genic (versus 38 intergenic) proviral integration sites identified across tumours were enriched in known cancer drivers using seven publicly available cancer driver lists (Supplementary Fig. 2). Indeed, under the selection model, genes interrupted by the provirus are expected to be enriched in cancer drivers (the affected gene, if any, is *a priori* more difficult to pinpoint for intergenic insertions). The enrichment was significant compared to random sets of genes matched for expression level in lymphocytes ($0.0016 < P < 0.0219$), strongly suggesting that—in contrast to the prevailing view—HTLV-1 and BLV integration sites are overrepresented in the vicinity of genes connected to cancer, and that the resulting perturbation is an essential driver of tumour formation (Table 1, Supplementary Data 3 and Supplementary Table 2).

**Provirus-dependent transcription interruption of host genes.** In 27 of the analysed tumours (11 in human and 16 in ruminants), the provirus was integrated in the intron of a gene with same transcriptional orientation (assuming dominant 5′LTR to 3′LTR viral transcription). We refer to this group as 'genic insertion, concordant'. In 21 of these, the RNA-seq data revealed premature termination of transcription and polyadenylation of the interrupted host gene at the viral poly-(A) signal (441 and 603 bp within the BLV and HTLV-1 5′LTR, respectively). This was accompanied by severe reduction of downstream exon reads ($P = 5.9e{-}08$, Mann–Whitney U-test), strongly suggesting *cis*-allele truncation of the affected host genes (Fig. 1). Expression of downstream exons was halved on average, suggesting that provirus-dependent transcriptional termination operates in all infected cells. In two of these tumours read numbers of downstream exons were even significantly <50% of controls, supporting additional perturbation of the *trans*-allele, as expected for tumour suppressor genes (tumours T1345 and M2532, 88% and 96% downregulation of *MSH2* (ref. 33) and *STARD7* (ref. 34), respectively; Fig. 1c). The list of truncated genes includes established anti-oncogenes such as *MSH2* and *BRCC3* (refs 33,35).

**Viral antisense RNA-dependent *cis*-perturbation of host genes.** Demonstrating 5′LTR-dependent transcriptional termination of the interrupted gene in 23% (21/92) of the tumours leaves the question open of what alternative molecular mechanisms perturb the presumed cancer drivers in the remaining 77% of tumours. To gain some insights into what these mechanisms might be, we carefully mined the RNA-seq data from the 71 (i.e., 92–21) remaining tumours. Mapping the RNA-seq reads to the proviral and host reference genomes revealed in all tumours the complete absence of

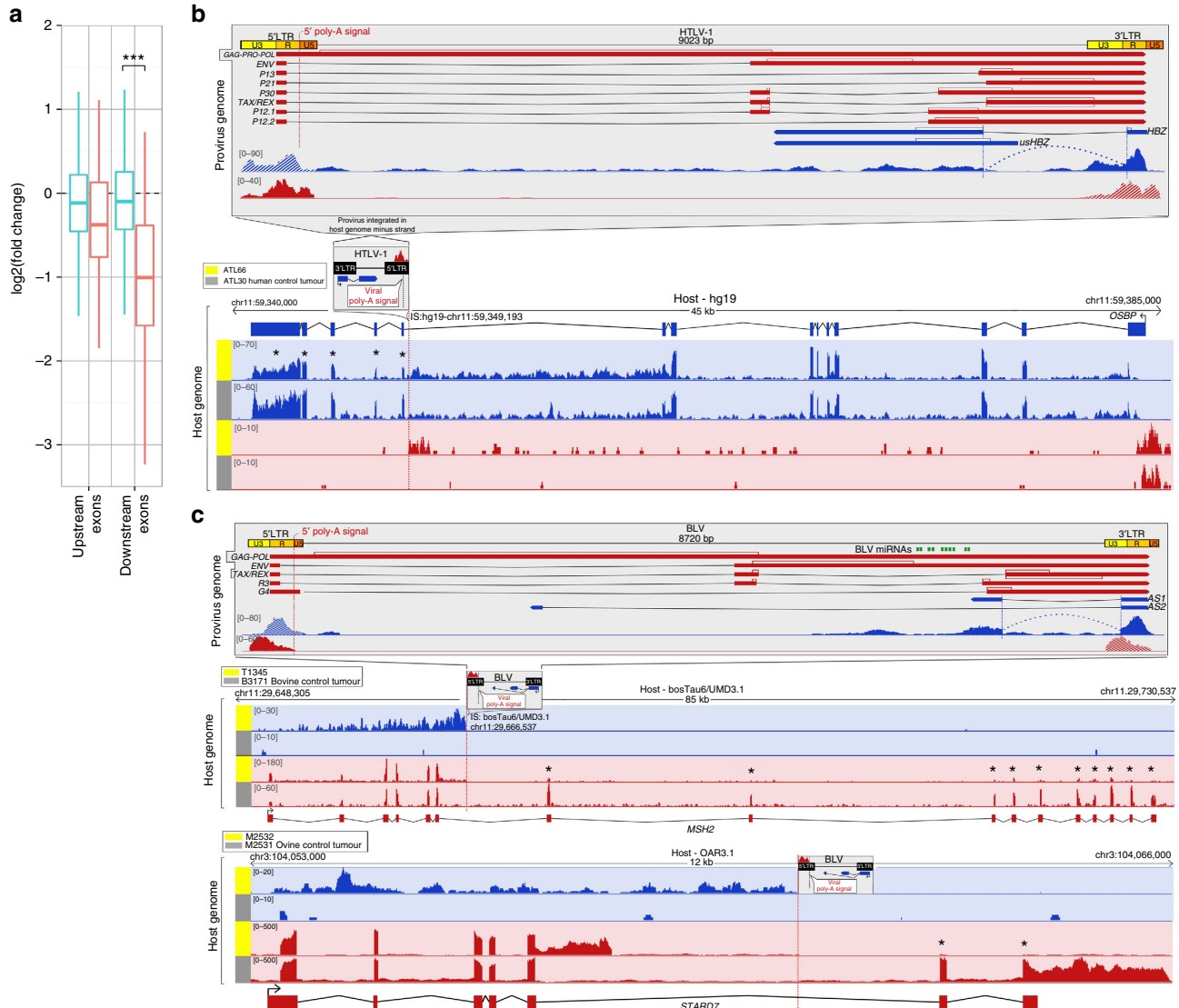

**Figure 1 | Provirus-dependent host gene interruption in HTLV-1/BLV primary tumours with genic-concordant proviral integration.** (**a**) Normalized RNA-seq read counts of upstream (left) and downstream (right) exons relative to the proviral integration site in the tumour set characterized by genic-concordant proviruses (red plot, $N = 21$) and control tumours without integration in that gene (blue plot). ***$P = 5.878e$-08 (Mann–Whitney $U$-test). (**b**) Transcription patterns of human leukaemia ATL66 shown as RNA-seq sense (red) and antisense (blue) coverage mapped to the proviral (top) or host (bottom) genomes visualized in Integrative Genomic Viewer (IGV)[59]. Top panel: HTLV-1 proviral genome flanked by 5′LTR/3′LTR redundant regions (U3, R, U5) that contain regulatory elements, transcriptional start sites (TSS) and poly-(A) signal. Positive-strand transcripts (red) encode structural and regulatory (TAX/REX) proteins; spliced *HBZ* antisense transcripts (blue) expressed from negative-strand. ATL66 RNA-seq coverage of HTLV-1: *HBZ* antisense transcripts and upstream coverage exposing hybrid transcripts; positive coverage of 5′LTR reveals read-through transcription and provirus-dependent premature polyadenylation of host gene *OSBP*. Absence of 5′LTR-driven viral transcription. Bottom panel: mapping to host genome (hg19). Small box: HTLV-1 integration in *OSBP* introns 9–10 (opposite orientation). *OSBP* exons 10–14 show decreased coverage (*ATL66/control ATLs ($N = 39$): 52% decrease, ATL66 *OSBP* downstream/upstream exons, fold-change = 0.52). Sense coverage: 3′LTR-dependent chimeric transcript in antisense overlap with *OSBP*. (**c**) Transcription patterns of bovine T1345/ovine M2532 B-cell tumours shown as RNA-seq sense (red) and antisense (blue) coverage mapped to the proviral (top) or host (bottom) genomes. Top: BLV genome, annotation and T1345 RNA-seq coverage representative of both tumours: *AS1* antisense transcription; positive coverage of 5′LTR reveals host gene transcription (read-through) and provirus-dependent premature polyadenylation. Bottom panels: mapping to host genomes (UMD3.1 and OAR3.1). Small box: BLV integration in *MSH2* (ref. 33) intron 6 or *STARD7* (ref. 34) intron 5. Decrease of *MSH2* and *STARD7* downstream exon coverage (*MSH2* T1345/control tumours ($N = 14$), 88% decrease; T1345 *MSH2* downstream/upstream exons, fold-change = 0.12 and *STARD7*: 96% decrease (control tumours, $N = 31$), downstream/upstream exon fold-change = 0.08). Antisense coverage: 3′AS-dependent chimeric transcript in antisense overlap with *MSH2/STARD7*. See also Supplementary Fig. 3 for RNA-seq coverage assignment to 5′LTR/3′LTR.

viral 5′LTR-dependent sense transcripts (5′S) (corresponding to the *GAG*, *POL*, *ENV* structural genes and the regulatory genes including *TAX*), yet abundant 3′LTR-dependent antisense transcripts corresponding to the previously described HTLV-1 *HBZ* and BLV *AS1/2* RNAs[11,22,24] (Fig. 2a and Supplementary Fig. 3).

Most importantly, it revealed the systematic interactions between the antisense transcripts (3′AS) and host genes located upstream of the provirus. (Fig. 2b, Supplementary Fig. 4 and Supplementary Table 3). These interactions conformed to four major schemes: capture of host exons located upstream of the provirus by the first

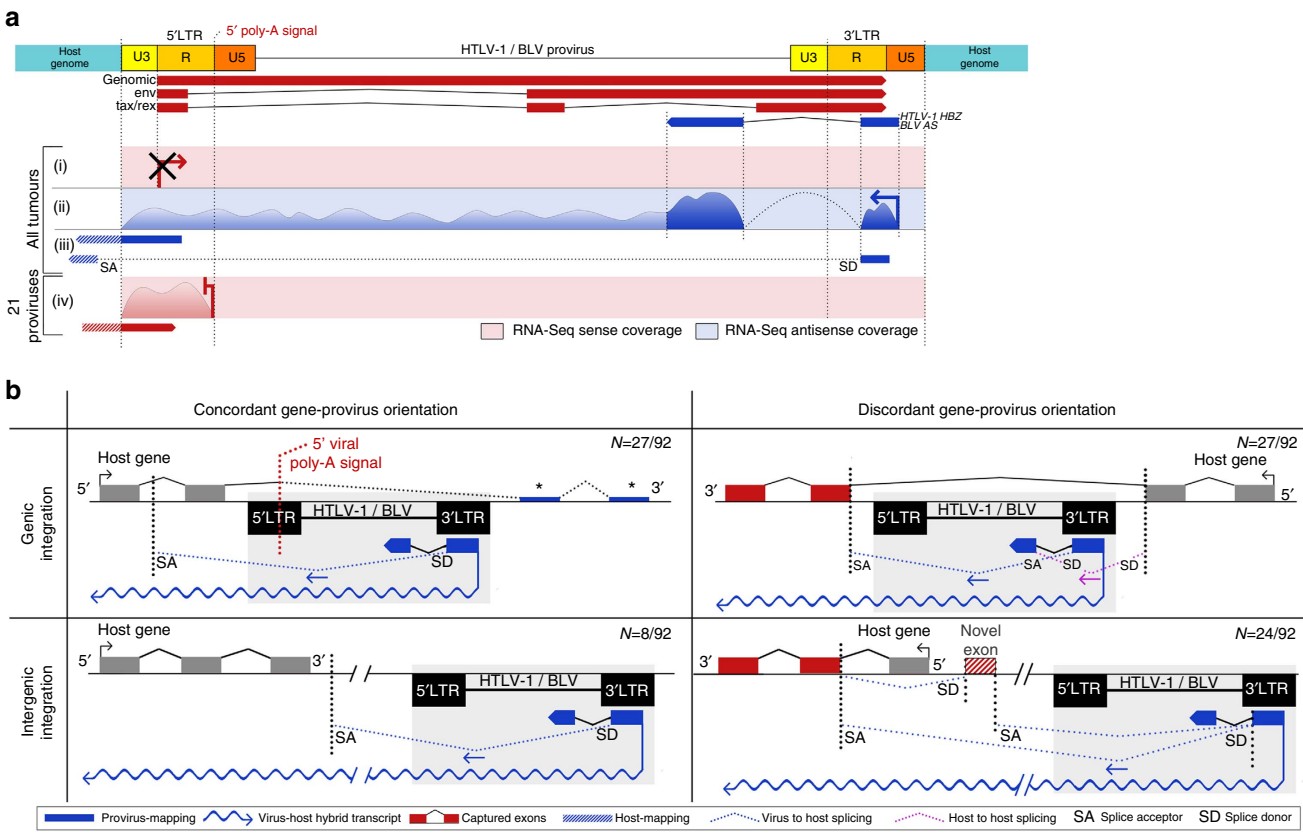

**Figure 2 | Viral antisense RNA-dependent transcriptional interactions with the host genome in HTLV-1/BLV primary tumours.** (**a**) Schematics of RNA-seq coverage mapped to HTLV-1/BLV in ATLs/B-cell tumours and antisense-predominant viral transcription. Top: proviral genome and simplified HTLV-1/BLV common annotation: positive-strand transcripts (red), spliced *HBZ/AS* antisense transcripts (blue). (i–iv): RNA-seq coverage. In all tumours, (i) absence of 5′LTR-dependent positive-strand coverage (structural proteins and TAX, Supplementary Fig. 3), (ii) 3′LTR-dependent *HBZ/AS* antisense transcripts (dark-blue) and upstream coverage (light-blue), (iii) hybrid antisense reads that span *HBZ/AS* exon 1-host and 5′LTR-host boundaries, supporting 3′LTR-dependent chimeric transcripts. In 21 proviruses (genic-concordant), (iv) positive coverage of host-5′LTR-U3/R boundary (viral poly-(A)-dependent host transcript truncation). See also Supplementary Fig. 3: coverage for 24 representative tumours and Supplementary Fig. 4: secondary types of virus-host transcriptional interactions. (**b**) Four main patterns of viral antisense RNA-dependent transcriptional interactions with the tumour genome: upper left: genic integration, concordant gene-provirus transcriptional orientation. Viral 5′LTR poly-(A)-dependent gene interruption, downstream exon decreased expression (*blue boxes, see also Fig. 1) and 3′AS-dependent hybrid transcript in antisense overlap with upstream sequences. Splicing to host cryptic SA (10/21 proviruses). Upper right: genic integration, discordant gene-provirus transcriptional orientation. 3′AS-dependent virus–host hybrid transcript and *HBZ/AS* exon 1 SD sequestration by upstream host exon(s) (i.e., tumour M160/*ICA1*; Fig. 3a). Capture of viral *HBZ/AS* exon 2 by downstream host gene exon (10/27 proviruses). Bottom left: intergenic integration, concordant gene-provirus transcriptional orientation. 3′AS-dependent virus–host chimeric transcript in antisense overlap with host gene (i.e., ATL1_Ly/*RGCC*; Supplementary Fig. 5a). Bottom right: intergenic integration, discordant gene–provirus transcriptional orientation. 3′AS-dependent chimeric transcript in sense overlap with upstream host gene(s) (i.e., tumours M138, M251 and M21/*FOXR2* and *RRAGB*; Fig. 3b,c), and capture of exon 2 or novel exon that creates non-canonical isoforms (i.e., tumour LB120/*SEPT11*; Fig. 3d). Six proviruses were integrated in gene deserts (Supplementary Table 1 and Supplementary Data 2). See Supplementary Figs 4 and 6: comprehensive characterization of dominant and secondary types of virus-host interactions in tumours. RNA-seq splice junctions/breakpoints were validated by RT–PCR for representative tumours of each group (Supplementary Fig. 7).

*HBZ/AS* exon (genic (27/71 tumours) and intergenic (12/71 tumours), discordant), antisense overlap of genes located upstream of the provirus by long 3′LTR-dependent antisense transcripts (genic (6/71 tumours) and intergenic (8/71 tumours) concordant), sense overlap of genes located upstream of the provirus by long 3′LTR-dependent antisense transcripts (23/71 tumours, intergenic discordant) and the capture of viral *HBZ/AS* exon 2 by host gene transcripts (10/27 tumours, genic-discordant) (Figs 2b and 3 and Supplementary Fig. 5). The genuine nature of the chimeric transcripts detected by RNA-seq was tested by RT–PCR (7 transcripts) or 3′ modified RACE (2 transcripts) and confirmed for all of these (Fig. 3c and Supplementary Fig. 7).

At least one such antisense RNA-dependent interaction was observed in every one of the 71 tumours. Across all 71 tumours, a total of 92 genes were involved, including 60 coding and 32 non-coding genes (Supplementary Data 2 and 4 and Supplementary Table 4). The distance between the perturbed host gene and the interacting provirus averaged 172 kb, ranging from zero in the cases of genic insertions to 1,300 kb in a case of intergenic insertion. Interactions were observed with a single (45 integrations) or multiple host genes per provirus (15 with 2, 3 with 3, and 2 with 4 genes). In 21 cases, the interacting host gene was not the gene most closely located to the provirus. Like these 71 proviruses, the 21 genic-concordant proviruses described above all showed evidence of 3′LTR dependent transcription with antisense overlap of upstream sequences in addition to the poly-(A) dependent interruption type of perturbation (Figs 1 and 2b, top left panel). In 5 of these 21 tumours, we observed additional interactions of this 3′LTR-dependent antisense transcript with the adjacent upstream

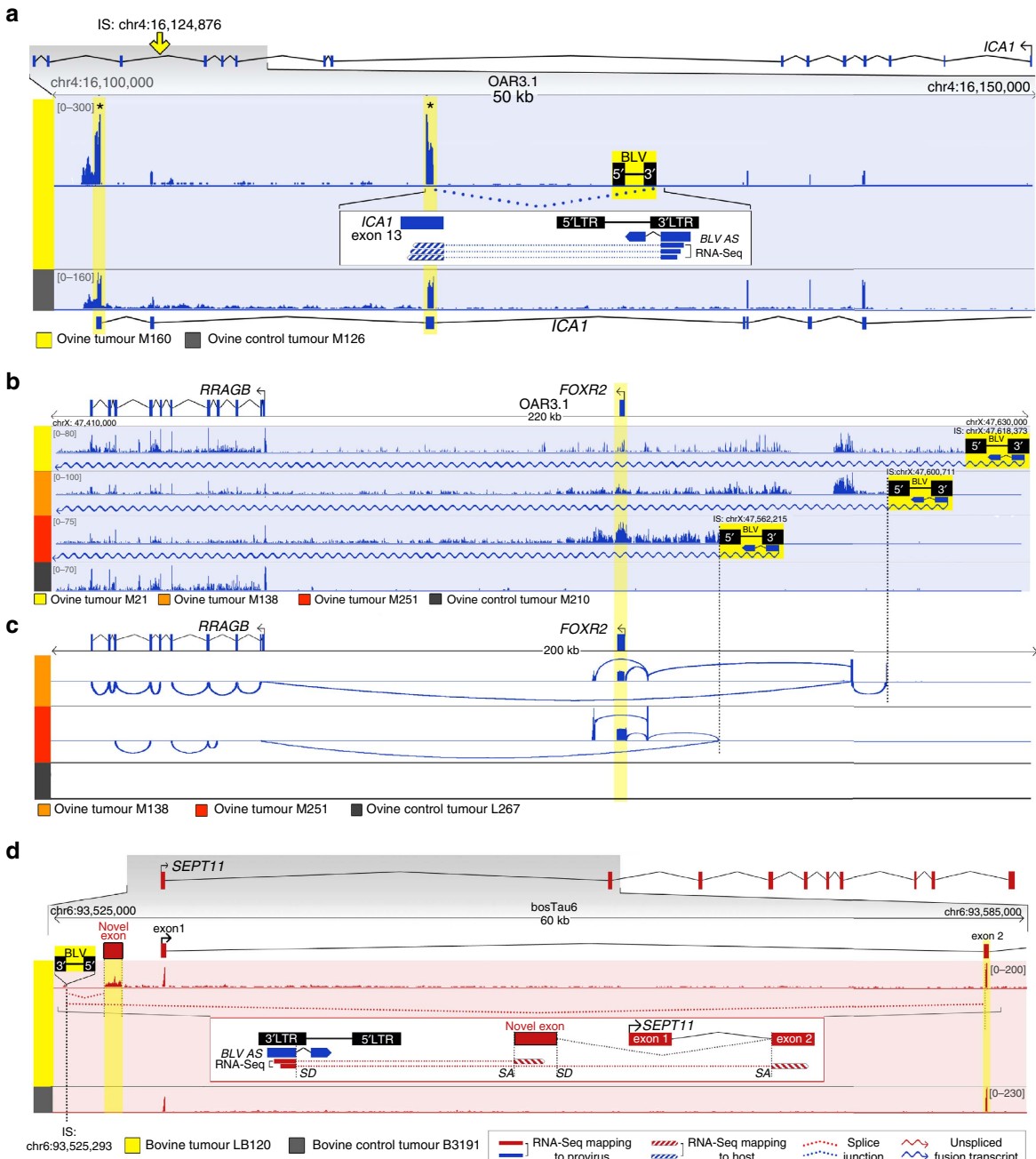

**Figure 3 | Viral antisense RNA-dependent *cis*-perturbation of host genes in representative tumours with discordant proviruses.** RNA-seq antisense (blue) and sense (red) coverages of tumours with genic (**a**) and intergenic (**b**–**d**) discordant proviruses. Upper (yellow) and lower (black) IGV tracks: tumour of interest and control tumour, respectively. (**a**) Genic-discordant provirus and host gene *cis*-perturbation by 3′AS-dependent capture of upstream exons. Ovine tumour M160: capture and increased coverage of *ICA1* exons 13–15; *read counts M160 *ICA1* upstream/downstream exons fold-change = 11.54; *ICA1* upstream exons M160/control tumours (N = 31) fold-change = 6.89. Box: hybrid RNA-seq split reads spanning BLV *AS* exon 1 and *ICA1* exon 13. (**b**) Intergenic-discordant provirus and interaction with multiple host genes: RNA-seq of three independent ovine tumours M251, M138, M21 with BLV integration upstream of *FOXR2* (ref. 42) (80 kb-window) reveals sense overlap of *FOXR2* and *RRAGB* (ref. 66) by 3′AS-dependent hybrid transcripts (160, 220 and 240 kb in length respectively). (**c**) 3′AS-capture RNA-seq reveals *AS* exon 1–*FOXR2*/*RRAGB* splice junctions and ectopic expression of *FOXR2*. Mapping and Sashimi plots of reads obtained from 3′AS-capture RNA libraries (STAR splice-aware aligner) prepared from modified 3′-RACE products of tumours M138 and M251 reveal *FOXR2* coverage as well as chimeric split reads exposing splicing events between BLV *AS* exon 1 and upstream genomic sequences including *FOXR2* and *RRAGB* (white tracks). BLV integration causes ectopic expression of *FOXR2* a gene that is not expressed in normal ovine B-cells (L267: control tumour), consistent with a gain-of-function of this well-established oncogene. (**d**) Intergenic-discordant proviruses generate novel transcript isoforms: host gene *cis*-perturbation by sense overlap of upstream gene by 3′AS-dependent hybrid transcript with capture of both exon 2 and a novel exon upstream of canonical exon 1, creating two novel isoforms (bovine tumour LB120: BLV integration upstream of *SEPT11* (ref. 67), *SEPT11* exon 1 skipping). See also Supplementary Fig. 5 for representative tumours with concordant proviruses.

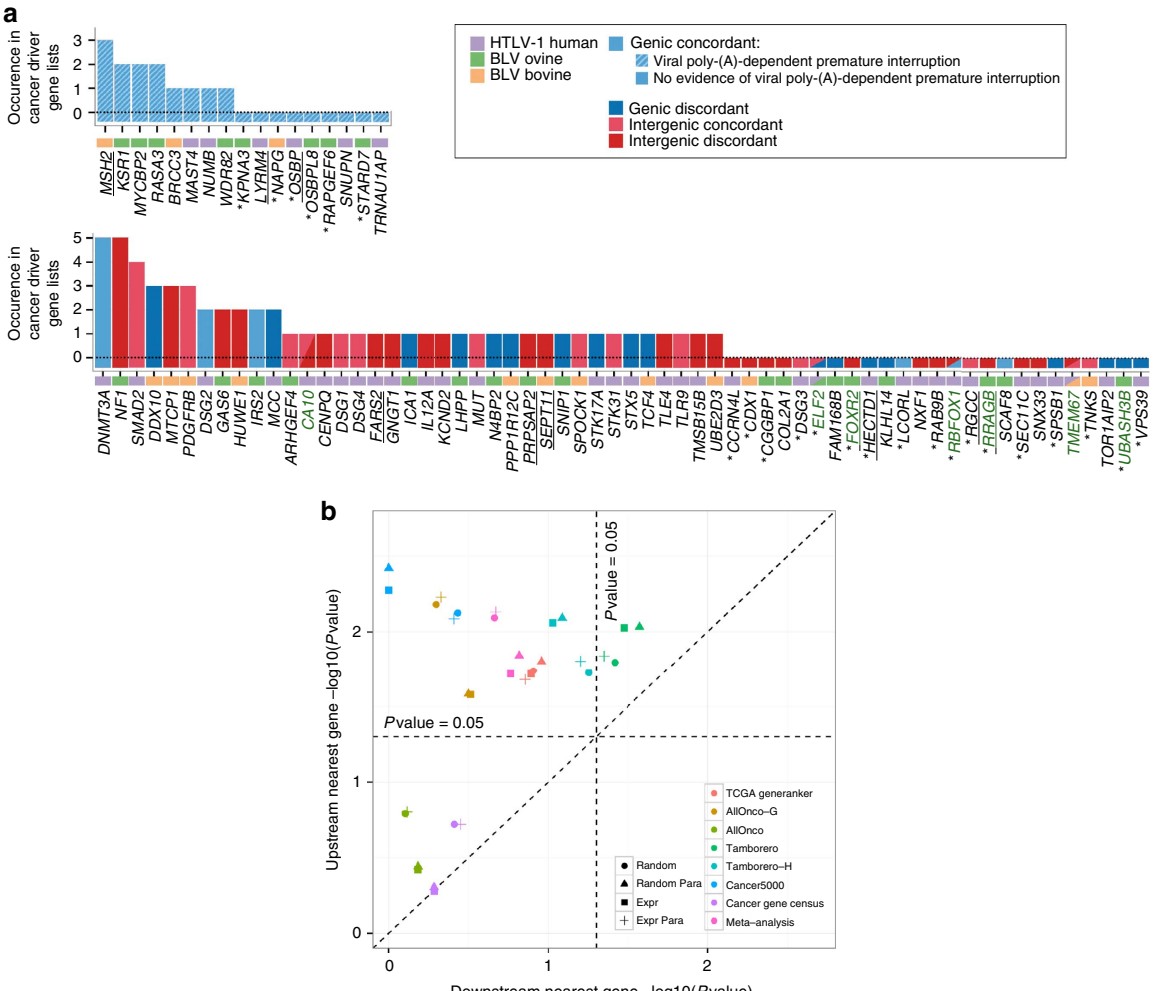

**Figure 4 | HTLV-1/BLV interacting host genes are connected to cancer.** (**a**) Transcript-interacting host genes were ranked according to their occurrence in seven cancer driver lists used for enrichment (Table 1 and Supplementary Table 2). Top panel: viral poly-(A) truncated genes disrupted by genic-concordant proviruses. Bottom panel: all genes interacting with 3'AS-dependent transcripts. Cancer drivers are equally represented between provirus types (genic, intergenic) and across species. Underlined genes: genes for which transcriptional patterns in tumours are shown in Figs 1 and 3 and Supplementary Fig. 5. Recurrent genes between tumours: green symbols. *Genes absent from cancer driver lists for which literature screen supported connection to cancer. This list comprises *FOXR2*, *RRAGB*, *ELF2* and *SPSB1* (refs 40,42,66,68), genes that exhibit undeniable oncogenic properties. *UBASH3B* (BLV, sheep): identified as the target gene of HTLV-1 integration in one of the ATLs analysed in a recent WGS study[11]. The remaining protein-coding genes and interacting non-coding RNAs not previously reported in cancer (Supplementary Data 2 and Supplementary Table 4, antisense transcript-interacting lncRNAs) represent a potential resource of novel candidate cancer drivers of both the coding and noncoding class of genes. (**b**) Host genes upstream (*y*-axis) of proviral integration sites in ATLs and B-cell tumours (92 sites) show significant enrichment in cancer drivers in contrast to the corresponding downstream host genes (*x*-axis), supporting antisense-dependent cancer driver perturbation by HTLV-1/BLV proviruses. The direct target genes of genic proviruses were excluded from the analysis. The significance of the enrichment was computed for seven publicly available cancer driver lists and for all list combined by means of a meta-analysis (Supplementary Data 3 and Supplementary Table 2). Observed scores were compared to simulated scores obtained from $N = 100,000$ size-matched random or expression-matched gene sets, including information about paralogs (Random Para, Expr Para) or not (Random, Expr) (Supplementary Fig. 2). Symbol code: simulated gene sets, colour code: cancer driver lists.

host gene, either by antisense overlap (2 tumours) or by exon capture (3 tumours; Supplementary Fig. 5b).

**Provirus-interacting genes are enriched in cancer drivers.** To test the biological relevance of the observed antisense RNA-dependent interactions, we examined the enrichment of the 65 (60 + 5) antisense interacting protein-coding genes in known cancer drivers (gene subsets and criteria for gene inclusion; Supplementary Data 3 and Supplementary Table 4). The enrichment was highly significant and robust, and dependent on both the 33 genic and 38 intergenic insertion sites of the three

species ($1e-05 < P < 0.0006$; Table 1 and Supplementary Data 3). Interacting genes included well-established cancer-connected genes such as *DNMT3A*, *SMAD2*, *MTCP1* and *TLE4* (refs 36–39) (Fig. 4a). Further supporting a role for viral antisense RNA-dependent interaction in tumorigenesis, we observed a significant enrichment in cancer drivers for the 43 nearest protein-coding genes located in a 1 Mb-window upstream of intergenic proviral integration sites while the matched list of 48 genes located downstream of these integration sites was not enriched (Fig. 4b). Note that Cook *et al.* (2014) presented evidence for loose association of HTLV-1 integration sites with Gene Ontology terms (cell morphology, immune cell trafficking,

haematological system development and function), in only 6% of the ATL cases examined (11/197) and only when restricting the analyses to genes located downstream but not upstream of the provirus (contrary to our findings which point upstream genes, consistent with antisense-dependent interaction). The reasons for these apparent discrepancies between their and our findings are not known.

Proviral antisense-dependent exon capture has the potential to cause expression of non-canonical isoforms of expressed genes, or ectopic expression of genes that are normally silent in the lymphoid lineage. Intriguing examples include N-terminal truncated isoforms of *ELF2* and *TCF4* (members of the *Ets* family of transcription factors and DNA binding transcriptional regulators of the *Wnt* pathway, respectively)[40,41], and ectopic expression of the well-established oncogene *FOXR2* (refs 42,43) in three independent ovine tumours (Fig. 3b,c). The capture of viral *HBZ/AS* exon 2 by host gene transcripts (10/27 genic-discordant cases) may cause premature transcription termination of the host gene at the *HBZ/AS* exon 2 poly (A) site (Fig. 2b, top right panel and Supplementary Fig. 4, scheme v). This suggests that genic-discordant insertions have the potential to affect the same gene by two distinct mechanisms—transcript interruption and downstream exon capture—consistent with observations in *Sleeping Beauty* transposon induced tumours in mice[44]. For the majority of intergenic proviruses (26/38), standard RNA-seq revealed sense or antisense gene overlap by provirus-dependent hybrid transcripts, yet without direct evidence of exon capture. Nevertheless, enrichment in cancer drivers for the 30 corresponding protein-coding genes was robust and highly significant ($0.0015 < P < 0.0004$, Table 1 and Supplementary Data 3), suggesting that these antisense-driven transcripts have a functional impact on the corresponding genes despite the lack of obvious transcriptional effects.

The strong bias for upstream interaction is in agreement with the absence of 5'LTR dependent mRNA transcription from the proviral positive-strand in all tumours examined, consistent with previous reports that showed antisense-predominant HTLV-1/BLV transcription and 5'LTR epigenetic silencing or *TAX* mutations in tumours[11,18–20,24] (Fig. 2a and Supplementary Fig. 3). It is noteworthy that we observed 3'LTR-driven antisense-dependent chimeric transcripts involving well-established cancer drivers located upstream of the provirus in the 11 ATLs with 5'LTR-deleted defective HTLV-1 proviruses (i.e., *SPSB1*; Supplementary Fig. 8). This strongly suggests that in fully malignant clones, positive-strand—and paradoxically *TAX*—silencing accompanies HTLV-1/BLV antisense-dependent host gene *cis*-perturbation presumably allowing the malignant clone to proliferate under strong host immune control[21].

Note that Kataoka *et al.* also reported HTLV-1-dependent read-through transcripts in ATLs. From 53 integration sites, the authors identified 12 aberrantly spliced fusions with the host genome, which all were produced from genic proviruses ($N = 23$). They did not report interactions with host genes in the case of intergenic proviruses ($N = 30$). While the authors conclude that the relevance of aberrant transcripts observed in ATL is unknown, our findings show that they tend to occur with genes that are enriched in known cancer drivers and hence play an important role in tumorigenesis.

Altogether, our findings support the notion that *cis*-perturbation of cancer drivers by the HTLV-1/BLV proviruses is an essential component of leukemogenesis.

**Non-random provirus distribution at asymptomatic stages**. The results reported thus far were obtained on late-stage tumours. They do not discriminate between a role for proviral integration

in promoting early-stage polyclonal expansion rather than late-stage precipitation to full-blown monoclonal tumour. To discriminate between these two hypotheses, we utilized the BLV experimental model in sheep. A considerable advantage of this model is that—contrary to the natural diseases in human and cattle—all infected sheep develop leukaemia/lymphoma, tumour onset occurs within a much shorter time frame (20 months on average) and it is possible to monitor infected animals at the very early stages of infection. We first comprehensively analysed proviral integration sites at early nonmalignant stages (characterized by the presence of multiple clones of low abundance). This was achieved by very deep, high-throughput DNA-seq-based mapping of BLV integration sites for 10 infected but still asymptomatic sheep (proviral load range: 0.02–34%, clone abundance range: 0.002–9.524%; Supplementary Data 1). It uncovered 66,557 unique integration events. Examining their chromosomal distribution revealed extreme non-randomness, defining 674 genic and 48 intergenic hotspots of integration (genome-wide corrected $P < 0.05$) (Fig. 5). We showed by simulation that the majority of genic hotspots could not be explained by expression level and gene size alone (false discovery rate (FDR) $< 0.1$ for 468/674 genic hotspots; proportion of true alternative (i.e., not explainable by expression level and size alone) hypotheses ($\pi_1$) $= 0.67$). The average number of integration sites per hotspot was 33 (range: 12–322) for genic hotspots and 37 (range: 12–202) for intergenic hotspots. The average size was 67,180 bp (range: 11,180–302,000) and 80,570 bp (range: 23,040–504,200) for genic and intergenic hotspots, respectively. Genes involved in genic hotspots showed a highly significant enrichment in cancer drivers ($P < 1e-5$; Supplementary Data 3) and a robust overlap with the 74 3'AS-interacting host genes identified in the HTLV-1/BLV tumour set ($P = 0.00073$; Fig. 5b and Supplementary Table 5). The list of genic hotspots includes established cancer drivers such as *ARID1A*, *ARID1B*, *CBL-B*, *PIK3CA* and *PTEN*[45–48] (Fig. 5c). Most interestingly, we observed a very strong signature of non-randomness of proviral orientation within hotspots (Fig. 5d–f). Of the 674 (76%) genic integration hotspots, 517 showed a significant bias (FDR $< 0.1$) towards either concordant (350) or discordant (167) orientation (with regard to the orientation of the host gene as defined above; concordant: provirus and gene in the same orientation; discordant: provirus and gene in opposite orientation). The genic hotspots with predominant concordant integration most probably point towards events of insertional inactivation by transcriptional termination, while hotspots with predominant discordant integration may point towards viral antisense-dependent exon captures. This finding certainly weakens the alternative (versus selection) hypothesis of chromatin-feature-dependent integration (which is unlikely to be orientation-dependent). Equally interesting, we observed the same orientation bias for non-genic integrations. Of the 48, 38 (79%) non-genic integration hotspots showed an orientation bias with FDR $< 0.1$ (see also Fig. 5e,f). This observation is in agreement with our hypothesis of BLV antisense RNA-dependent perturbation of cancer driver genes.

**Virus–host hybrid-RNA signatures at asymptomatic stages**. To further test this hypothesis, we followed up on this DNA-seq-based study and performed RNA-seq of BLV antisense-enriched RNA from the same asymptomatic animals (3'AS-capture RNA-seq). This revealed BLV 3'AS-host chimeric transcripts involving multiple genes per sample (range of 4–276, mean of 84; Supplementary Table 3), supporting a total of 723 interacting host genes. Genes involved showed a highly significant level of

recurrent capture of the same host genes between asymptomatic individuals ($P < 1e-5$), a significant overlap with transcript-interacting host genes previously identified in malignant clones of the three species ($P = 0.00085$; Fig. 5b and Supplementary Table 6), and a highly significant enrichment in cancer drivers ($P < 1e-05$; Table 1 and Supplementary Data 3). As expected, we

observed a very strong overlap between the genes exposed by 3′AS-capture RNA-seq hybrid reads and the genic hotspots identified by DNA-seq mapping of integration sites in the same asymptomatic individuals ($P < 1e-5$; Fig. 5b). Most interestingly, the 3′AS-capture RNA-seq data revealed mapping of viral antisense RNA-host chimeric reads upstream of intergenic

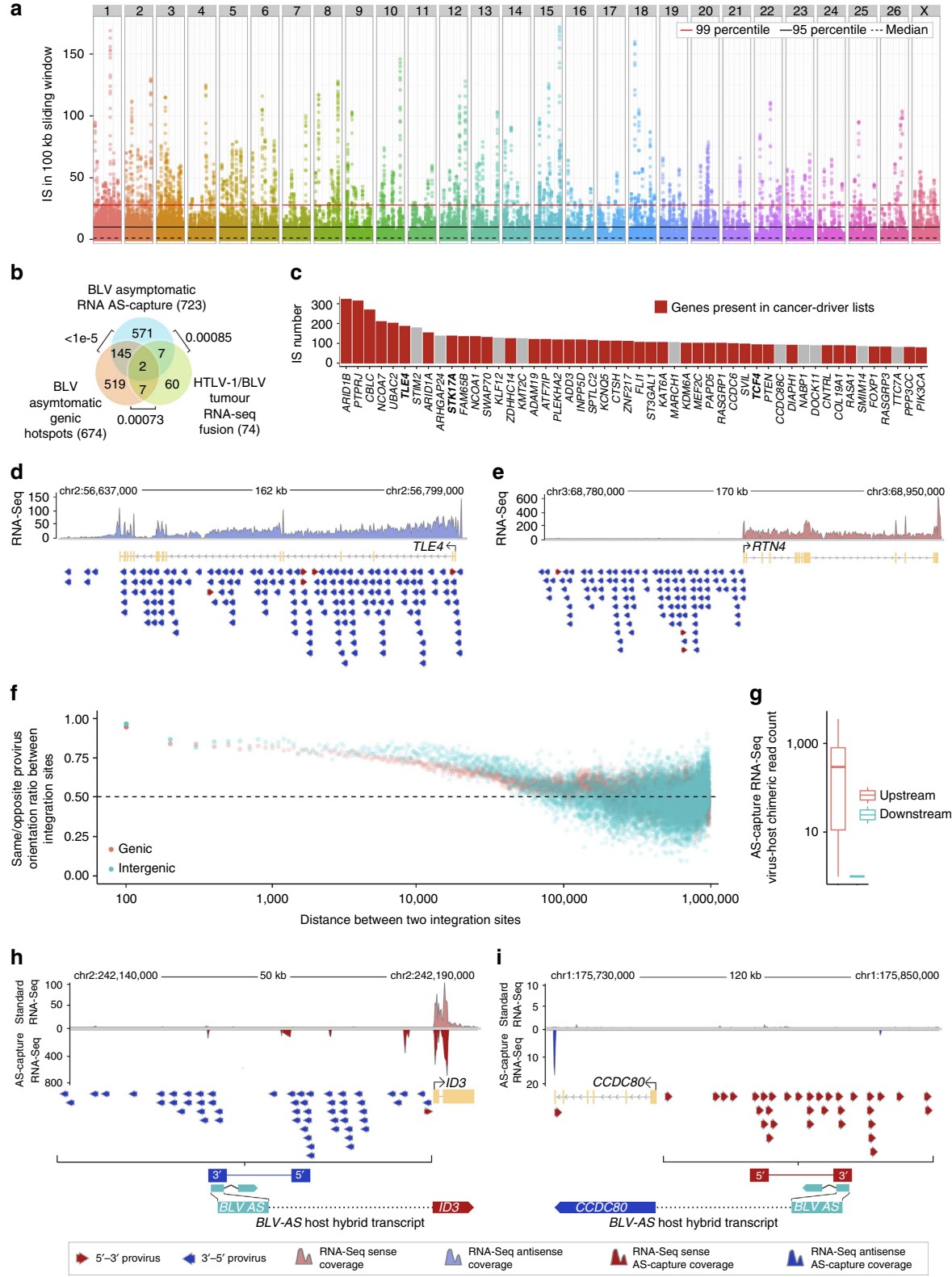

hotspots defined by DNA-seq (Fig. 5g). It also uncovered typical splice junctions between the BLV *AS* transcript and the corresponding upstream gene. This revealed cases of potential activation of important oncogenes (i.e., *ID3* (ref. 49), Fig. 5h), and included other examples (in addition to *FOXR2*) that may lead to ectopic expression of otherwise silent genes (i.e., *CCDC80*; Fig. 5i). These observations are in perfect agreement with our hypothesis of viral antisense RNA-dependent perturbation of cancer drivers.

**Assessing integration hotspots and cancer driver enrichment.** When infected animals develop full-blown leukaemia following the asymptomatic polyclonal phase, the tumour clone is assumed to expand and dominate, yet coexist with multiple infected clones that remain in the background. Further mining the data from the DNA-seq-based integration mapping of 22/32 ovine tumours indeed revealed 8,015 minor proviral integration sites in addition to the 22 dominant ones described above. As expected, this novel collection of integration sites was characterized by highly significant non-randomness, defining 65 hotspots (61 genic and 4 intergenic) that all overlapped with the hotspots defined in the asymptomatic samples. Forty-five (42 genic and 3 intergenic, 70%) of the hotspots were characterized by significant (FDR ≤ 0.10) orientation bias (33 concordant, 9 discordant for genic hotspots). Also as expected the genes corresponding to the genic hotspots were highly enriched in cancer drivers ($P < 1e-5$). The same approach was then applied to 31/44 human ATL cases. This allowed for the identification of 4,628 minor proviral integration sites in addition to the 32 tumour-specific ones described above. Surprisingly, examining their chromosomal distribution did not reveal any evidence for integration hotspots or orientation bias. We expanded our collection of HTLV-1 proviral integration sites with a recently published data set of 11,279 sites[10,50] to reach a total of 15,939, but this did not alter the outcome regarding integration hotspots or orientation bias. These findings are in agreement with results published by others[10,11,28]. Paradoxically, the 7,155 human genic integration sites were very significantly biased towards known cancer driver genes ($P < 1e-5$), genic hotspots defined in sheep ($P = 2.88e-13$) and *cis*-perturbed host genes previously identified in malignant clones of the three species ($P = 6.68e-05$) (Supplementary Tables 7 and 8). To understand the apparently discrepant results in the human samples, i.e., lack of evidence for integration hotspots yet strong enrichment of genic integration sites in cancer drivers and genic

hotspots defined in sheep, we performed simulations assuming that a fraction $w$ of $x$ proviral integration sites are sampled from clones that are undergoing expansion due to perturbation of one of $y$ cancer drivers (out of a total of 20,000 genes), of which a fraction $z$ is reported in cancer driver lists. The remaining fraction $(1 - w)$ of proviral integration sites are assumed to be sampled from infected but non-expanded leukocyte clones. $w$ was varied from 0.01 to 1, $x$ from 1,000 to 100,000, $y$ from 500 to 3,000 and $z$ from 0.25 to 1. It was obvious that the statistical test for enrichment in cancer drivers was considerably more powerful than that for the detection of integration hotspots for a substantial proportion of parameter space (Supplementary Fig. 10). This is mostly due to the requirement to adjust the hotspot test but not the enrichment test for multiple testing. Particularly noteworthy was the effect of a decreasing proportion of integration sites sampled from expanded clones ($w$). This fundamental power difference between the two tests may thus very well explain the apparent discrepancy observed with the human data, and the fact that integration hotspots were not reported before. The model used in the simulations, characterized by two distinct populations of infected lymphocytes (one expanding with proviral insertions affecting cancer drivers, and one not expanding with random proviral insertions), predicts that the degree of integration in hotspots and of enrichment in cancer drivers should be correlated with corresponding clonal abundance (as expanding clones are by definition more abundant). We tested this prediction using the large number of insertion sites available in asymptomatic sheep and found that it was indeed the case (Supplementary Table 9).

### Discussion
Taken together, our results strongly support the notion that *cis*-perturbation of cancer drivers by the provirus is a major determinant of early clonal expansion in both BLV and HTLV-1 induced leukaemia. We provide circumstantiated evidence that the absence of easily detectable integration hotspots yet enrichment in cancer drivers in the human natural host (contrary to the flagrant hotspots detected in the ovine model) may reflect species-specific dynamics of infected-expanding versus infected-nonexpanding lymphocyte populations and hence proportions. This may point towards disparities in the antiviral immune response—a major driving force underlying clone abundance in HTLV-1 individuals[21,28]—between the experimental model and the human disease. It is tempting to speculate that this may also

**Figure 5 | Hotspots of proviral integration at polyclonal nonmalignant stages of infection.** (**a**) Genome-wide distribution of BLV integration sites in asymptomatic sheep samples. Y-axis: number of integration sites per genomic bin (100 kb overlapping genomic windows sliding by steps of 50 kb). Hotspots of proviral integration were identified by simulation (Methods), defining 674 genic and 48 intergenic hotspots ($P < 0.05$). (**b**) Significant recurrence ($P$-values) between genes revealed by BLV genic integration hotspots (674 genes), antisense-RNA interacting genes identified in tumours (74 genes, HTLV-1/BLV) and genes identified by 3′AS-capture RNA-seq of asymptomatic samples (723 genes). All gene subsets showed robust cancer driver enrichment ($P < 1e-05$ and $7e-04$ for asymptomatic and tumour samples, respectively, Supplementary Data 3). (**c**) Top 50 genic integration hotspots. Comprise gene classes like chromatin modifiers, E3-ubiquitin ligases and tumour suppressors (*ARID1B*, *CBL-B*, *PTEN*). Genes in bold: also identified in tumour RNA-seq data set (*TLE4* and *STK17A*: ATLs, *TCF4*: bovine B-cell tumour). (**d**) Genic integration hotspot in tumour suppressor *TLE4* (ref. 39): arrows represent proviruses (5′–3′ orientation). Of 163 sites, 159 show identical orientation (ratio same/opposite: 0.97) consistent with genic-concordant proviruses predicted to cause *TLE4* loss-of-function. *TLE4* also affected in tumour data set (ATL62_2). (**e**) Intergenic integration hotspot upstream of *RTN4* (ref. 69): 121/124 sites show identical orientation (ratio same/opposite: 0.97) consistent with intergenic-discordant proviruses predicted to cause *RTN4* activation (gain-of-function). Mixed genic–intergenic hotspot type shown in Supplementary Fig. 9 (*ATF7IP*). (**f**) Provirus pairs from genic (red) and intergenic (blue) IS data sets were scored for relative orientation and same/opposite ratios computed for all combinations of pairs (Methods). Bias towards same orientation is associated with provirus proximity, consistent with non-randomness of proviral orientation in hotspots. (**g**) Virus–host chimeric transcripts uncovered by 3′AS-capture RNA-seq map to genomic region upstream of intergenic hotspots consistent with antisense-dependent transcriptional activity. Absence of coverage for corresponding downstream regions. (**h**) Mapping of 3′AS-capture RNA-seq hybrid reads (red coverage) to genomic region upstream of intergenic hotspot chr2: 242,107,500-242,240,229 reveals antisense-dependent chimeric transcription and interaction with oncogene *ID3* (ref. 49). (**i**) Hybrid reads mapping to genomic region upstream of intergenic hotspot chr1: 175,765,639–175,927,608 (blue coverage) reveal ectopic expression of *CCDC80* (not expressed in lymphocytes) consistent with a gain-of-function mechanism.

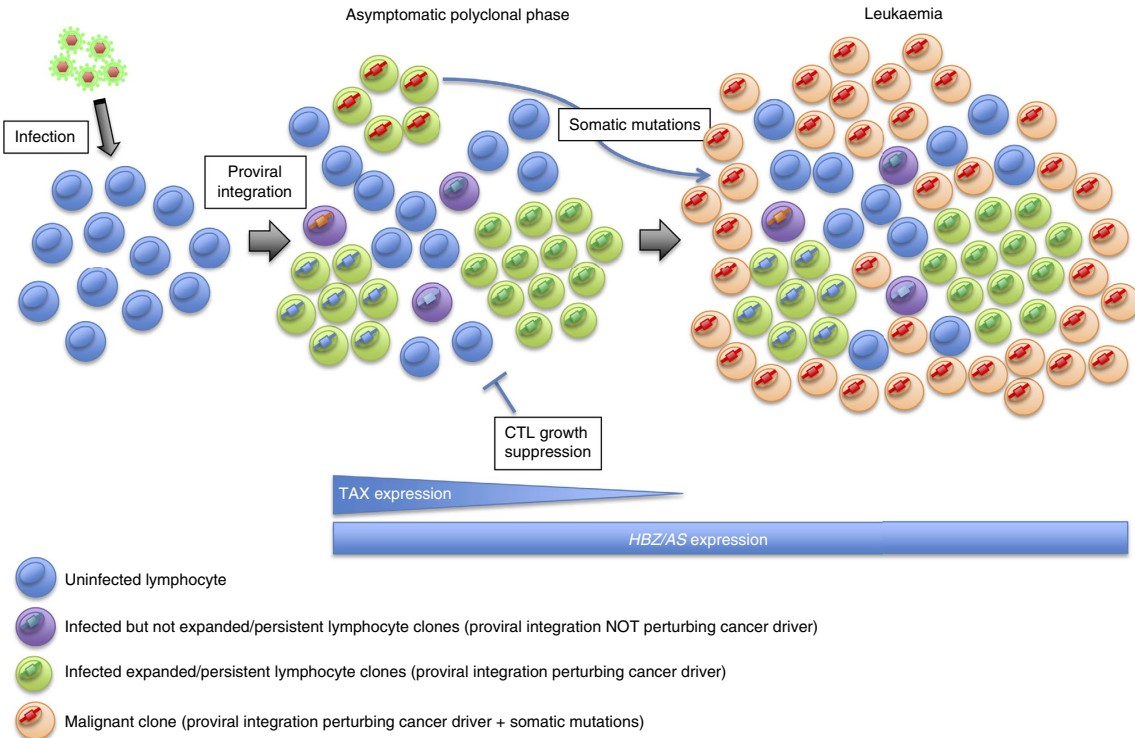

**Figure 6 | Model of leukemogenesis by HTLV-1/BLV.** After infection by HTLV-1/BLV the fate of a given infected T-cell/B-cell clone depends on the proviral integration site within the host genome, the expression of TAX and HTLV-1-*HBZ*/BLV-*AS*, the host CTL response to HTLV-1/BLV antigens and somatic mutations in the host genome. Asymptomatic polyclonal stage: the integration of HTLV-1/BLV proviruses in the vicinity of cancer drivers causes their perturbation and hence favours the persistence/survival/expansion of the corresponding infected clone (green clones). Clones in which proviral insertions do not affect cancer drivers show a modest survival (purple clones). The relative contribution of each infected clone to the polyclonal population of infected cells results from the balance between cancer driver perturbation, expression of TAX and *HBZ/AS* that both promote cell growth, and negative selection by the host CTL response. The prolonged life-span of clones in which cancer drivers are perturbed favours the acquisition of further somatic alterations in the host genome. Malignant stage: the accumulation of somatic changes ultimately precipitates the progression of one of the clones to full-blown malignancy (green clone—red integration and orange leukaemic clone). The tumour clone originates from an expanded/persistent clone yet not necessarily the most abundant one. The absence of TAX expression in the tumour clone confers a survival advantage through escape from the strong CTL immune response.

underlie the observed species-specific latency time. For example, if the half-life of infected-expanding lymphocytes was more drastically increased in sheep than in human and bovine, it might explain their higher proportion among infected lymphocytes (and hence increased power to detect integration hotpots) and reduced latency time (due to the increased probability to acquire the somatic mutations needed for progression to full-blown cancer).

Combining DNA-seq and RNA-seq data, yields a list of 674 putative leukaemia driver genes of which 370 were reported in one or several cancer-related gene lists used for enrichment analysis. Of the 50 most significant ovine genic hotspots (Fig. 5c), 40 were reported in these lists. Six of the 10 remaining genes, despite not being listed, were genes for which literature search supported undeniable oncogenic properties (*STIM2, ARHGAP24, KLF12, KMT2C, CCDC88C* and *DOCK11*), and 4 genes were—to our knowledge—not previously reported in cancer, yet may include new candidate drivers. *MARCH1*, a gene that exhibits E3 ubiquitin ligase activity and was recently reported to regulate MHC class II turnover[51], is an example of such a candidate. Thus, the catalogue of genes revealed by proviral integration hotspot identification represents a potential resource of novel cancer drivers. It may be particularly attractive for the discovery of new cancer-related noncoding RNAs considering that several genes uncovered by this work belong to that class.

In conclusion, although there is considerable evidence from previous work that the viral products—TAX and HBZ/AS—and the acquisition of somatic alterations in the host genome play a critical role in tumour development, we herein uncover an additional previously unrecognized yet complementary mechanism that contributes to leukemogenesis. We demonstrate that in tumour clones the HTLV1/BLV proviruses are integrated in the vicinity of cancer driver genes which they affect by either premature transcription interruption or antisense dependent *cis*-perturbation. We show that the same pattern already exists at early asymptomatic stages of infection. Thus, *cis*-perturbation of key host genes may contribute to malignant progression by providing a polyclonal background of infected cells with increased survival or proliferation. This extended half-life will promote the accumulation of further secondary mutations in the rest of the genome, ultimately precipitating the progression of one of these clones to full-blown ATL/B-cell malignancy (Fig. 6). Our results suggest that pharmacological repression of 3′LTR-dependent transcription may lessen polyclonal expansion during the asymptomatic/chronic stage of the disease, thereby delaying the emergence of the tumour.

## Methods

**Samples.** Samples from HTLV-1-infected individuals were collected after informed consent obtained in accordance with the Declaration of Helsinki and after institutional review board-approved protocol at the Necker Hospital, University of Paris, France in accordance with the 'Comité d'éthique Ile de France II'. We selected human ATL samples for sequencing on the basis of the

availability of sufficient DNA/RNA of clinical samples available from the Necker Hospital at the time of initiation of the study ($N = 44$). Samples consisted of peripheral blood mononuclear cells (PBMCs) from 21 acute ATLs, 3 chronic ATLs, 3 lymphoma-affected patients and 7 acute ATL patients that underwent therapy. Six samples were collected from lymphoma (lymph node and skin). Samples from asymptomatic carriers (AC) comprised three blood samples and one sample from an AC lymph node. Control samples consisted of uninfected CD4+ T-cells. PBMCs were isolated from blood using Histopaque-1077 (Sigma). Primary leukaemic B-cells and lymphoid tumours from BLV-infected sheep (Suffolk and polled Dorset crossed with Arcott breeds of either sex; $N = 32$) were collected at the acute stage of the disease (latency prior to tumour development range of 15–48 months). Sheep were housed at the Centre de Recherches Vétérinaires et Agrochimiques (Brussels, Belgium) and at the Vaccine and Infectious Disease Organization (VIDO-Intervac, Saskatoon, Canada). Experimental procedures approved by the Comité d'Ethique Médicale de la Faculté de Médecine, ULB were conducted in accordance with national and institutional guidelines for animal care and use. Asymptomatic sheep samples came from animals infected with the molecular clone pBLV344[7] following experimental procedures approved by the University of Saskatchewan Animal Care Committee, following Canadian Council on Animal Care Guidelines (Protocol #19940212). PBMCs were isolated from EDTA-treated blood using standard Ficoll-Hypaque separation. Lymphoid tumours were minced through a nylon mesh cell strainer (Becton Dickinson) to obtain single-cell suspensions. B-cell percentages were measured by fluorescence-activated cell sorting (FACS) using CD72-specific mAb DU2-104 (VMRD) and were ≥93% for all malignant samples used for further investigation. Samples from asymptomatic BLV-infected sheep were collected between 7 weeks to 34 month following experimental inoculation. Ovine control samples consisted of PBMCs of uninfected sheep assigned to the control group using physical randomization at the time of experimental inoculation with BLV ($N = 3$). Bovine B-cell tumours ($N = 15$) taken from our tumour collection stored at $-80\,°C$ comprised samples from various geographical origins (Japan, France, United States, and Belgium). Tumour samples were collected from blood or B-cell lymphoid masses developing following natural BLV infection some of which have been characterized previously[52]. The latency period before tumour onset in these animals is not accurately documented. Bovine control samples consisted of PBMCs isolated from seronegative animals. The total animal tumour sample size ($N = 47$) was defined to match clinical sample numbers. No statistical test was used to determine adequate sample size. The study did not use blinding. Detailed HTLV-1 patients' and animal samples' information is available in Supplementary Data 1 and 2. YR2 and L267 are tumour B-cell lines derived from ovine B-cell tumours M395 (B-cell leukaemia) and T267 (B-cell lymphoma), respectively[17,18]. Cell cultures selected for RNA-seq were validated using FACS labelling as previously described and tested for the absence of mycoplasma contamination.

**RNA sequencing.** Total RNA was extracted using the Qiagen AllPrep DNA/RNA kit and strand-specific ribosomal RNA-depleted RNA-seq libraries were prepared using the Illumina TruSeq Total RNA stranded kit. Libraries were analysed on an Agilent Bioanalyser DNA 1000 and quantified by qPCR using the KAPA kit (KAPA Biosystems). Sequencing was carried out on an Illumina HiSeq 2000 ($2 \times 100$ bp paired-end reads) and generated $\sim 2 \times 60$ million raw paired-end reads per library. The RNA-seq raw reads were aligned using STAR (v2.3.1.u)[53] and custom host–provirus genome references build using host reference genomes hg19 (human), UMD3.1 (bovine) or OAR3.1 (ovine), respectively, and proviral genome sequences BLV-YR2 (GenBank: KT122858) and HTLV-1-ATK-1 (GenBank: J02029), respectively. SAMtools and BAMtools were used to sort and index, and separate sense and antisense reads from the STAR output, respectively[54,55]. ENSEMBL v84 was used for host genome annotations and custom annotations for both HTLV-1 and BLV genomes. FeatureCounts[56], R packages DESeq2 (ref. 57) and DEXSeq[58] were used for read quantification, normalization and differential gene and exon expression analysis, respectively. Integrative Genomic Viewer (IGV) was used for visualization of sequencing alignments on both the host and viral genomes[59]. RNA-seq reads stemming from any of the viral LTRs (HTLV-1 and BLV) systematically mapped to both the 5′ and 3′ LTR as both are identical in sequence. Reads were specifically assigned to one or the other LTR based on additional host-specific mapping data and fusion-read information (outlined below): upstream or downstream position of host-mapping mate pair alignment, 3′AS RNA-host hybrid read identification (antisense coverage, 3′LTR alignment), host-LTR hybrid read identification (sense coverage, 5′LTR alignment), LTR-host read identification (sense coverage, 3′LTR alignment), host genomic environment (genic, intergenic) and evidence of 5′LTR deletion from HTS-based integration mapping data.

**Detecting hybrid RNAs and insertion sites from RNA-seq data.** Mispaired and soft-clipped reads supporting virus–host hybrid transcripts were identified using a custom two-pass alignment scheme as described in ViralFusionSeq[60]. RNA-seq paired-end reads were aligned to the host genome using BWA[61] (default parameter except -k 19 and -L 1) and mispaired and soft-clipped reads (minimum 8 soft-clipped nts) were re-aligned to the proviral genome. Host and proviral alignments of each read were compared, and reads were flagged as fusion reads if one read mapped to the host genome and the other to the proviral genome, or if

their soft-clipped breakpoint was localized within a 5 nt window. Fusion reads sharing the same fusion point within a 5nt window were clustered. Fusions were classified as genic/intergenic and concordant/discordant using a combination of BEDTools 2.16.2 (ref. 62) and BEDOPS 2.30 (ref. 63). HTLV-1/BLV integration sites were determined based on identification of chimeric transcripts that encompassed LTR-host boundaries. Hybrid reads generated from 3′AS-enriched RNA-seq data (asymptomatic samples) were additionally filtered to exclude both tumour-specific chimeric reads and hybrid reads common to multiple asymptomatic samples to exclude artefacts from sample cross-contamination. RNA-seq read coverage, virus–host annotated junctions and hybrid reads were loaded onto IGV to visualize the viral and host-specific transcriptomes and assess the consequences of proviral integration/transcription on the host transcription patterns.

**3′AS-capture RNA-seq.** Using total RNA as template, cDNA was produced with SuperScript III Reverse Transcriptase (Life technologies) following the manufacturer's instruction and primed with an oligo-dT tailed with the Nextera reverse sequence attached (Integrated DNA Technologies). cDNA was treated with RNase H (New England Biolabs) and semi-nested PCR was carried out to enrich the BLV fusions. The first PCR was performed using primers LTR1 (matches BLV AS exon 1) and NexRs (matches end of the Nextera-oligo dT) and Q5 High-Fidelity DNA Polymerase (New England Biolabs) with an annealing temperature of 66 °C and a 4 min extension (25 cycles). In the second PCR the LTR1 primer was replaced by LTR2 and NexR was reused. The PCR product was sheared in a Bioruptor Pico (Diagenode) following the manufacturer's instructions for fragments of $\sim 400$ bp, treated with the NEBNext Ultra End Repair/dA-Tailing Module (New England Biolabs) and ligated to a Y adapter produced by annealing oligos corresponding to the Nextera forward and reverse sequences using T4 DNA Ligase (New England Biolabs). The resultant DNA was indexed with Nextera XT indexes (Illumina) and libraries were mixed in equal proportions. Sequencing was carried out on an Illumina MiSeq instrument with $2 \times 150$ bp reads (Reagent Kit v2). Primer sequences are available in Supplementary Data 5. Hybrid RNA-seq reads and chimeric transcripts were analysed as described above (standard RNA-seq). Hybrid reads upstream or downstream of intergenic hotspots were identified in genomic windows defined as the region spanning from the hotspot extremity to the nearest protein-coding gene.

**HTS integration mapping and measure of clonal abundance.** To identify HTLV-1 and BLV proviral integration sites we used a method similar to that outlined by Gillet[9,29], but with a number of key changes to increase sensitivity and reduce costs by simplifying multiplexing[24]. In the case of asymptomatic samples, we used a modified method that included an extension step using a Hot start Taq Polymerase (Promega) with 25% of the dTTPs replaced with Biotin-11-dUTP (Thermo Scientific) followed by streptavidin-based selection (Dynabeads M-280 Streptavidin, Invitrogen/Life Technologies), allowing a reduced number of PCR cycles and removing the need for End Repair/dA-Tailing (15 cycles, annealing temperature of 66 °C with a 30 s extension) prior to addition of Nextera XT indexes (Illumina). Primer sequences are available in Supplementary Data 5. Paired-end reads were aligned to a host–provirus hybrid genome using BWA. After quality trimming (average base quality ≥30) only paired-end reads that fulfilled the following conditions (spanning LTR-host junctions) were retained: Read 1: BLV 5′LTR: 30 nts, BLV 3′LTR: 27nts, HTLV-1 5′LTR: 29 nts, HTLV-1 3′LTR: 45 nts of the read mapped to the relevant LTR extremity. Read 2: the read mapped to the host genome with ≤3 mismatches. Duplicates were removed based on reads that showed the same genomic insertion site and identical eight random nt tags. Read numbers were counted for each proviral integration site and reported using in-house R and Perl scripts. Clone abundance in tumours was determined as follows: if both 5′ and 3′LTR flanking sites were identified, % = average 3′–5′LTR. If only one flanking site was detected: % = % defined by detected LTR flanking site and provirus identified as either LTR-deleted if deletion is supported by evidence from RNA-seq hybrid read detection (non-canonical virus–host boundary) or full-length if RNA-seq-based data support the presence of LTR-host fusion reads.

**Identifying hotspots of proviral integration.** Identical proviral integration sites (IS) across samples were removed to account for potential cross contamination between samples and the IS that showed the highest read count was retained. 5′ and 3′ LTR flanking IS with same proviral orientation located within a 10 bp window were merged. IS distribution across the genome was first examined by counting the number of IS in sliding consecutive bins of 100 kb (50 kb overlap) and bins showing IS numbers >99th percentile of the distribution of IS per bin ('hotbins') were visualized in individual chromosomes. Hotspots were then defined as follows (contrary to hotbins that have a predetermined size of 100 kb, hotspots defined by simulation show a range of sizes): for each IS we counted the number of IS located in a 50 kb window centred on that IS = IS(i). We then randomly picked 66,557 IS in the ovine genome, performed the same counting procedure and retained the simulated IS window harbouring the highest number of IS = max.simIS. We performed $N = 1,000,000$ iterations of this procedure to generate a distribution reflecting random integration. We then assigned a $P$-value to each IS: $P$-value IS(i) = # max simIS > = IS(i)/1,000,000. Hotspots were defined by grouping consecutive IS windows that showed a $P$-value ≤0.05, the position of the two

extreme IS ±5 kb determining the boundaries of the hotspot. Hotspots were defined as genic if the median of the IS in the hotspot fell within a gene. Intergenic hotspots defined by simulation were then manually curated to account for poor annotation of the ovine transcriptome (ENSEMBL OAR3.1 v84) and re-assigned to the genic class if they fell within transcribed regions (i.e., non annotated exon, 5′ transcribed region of a mis-annotated gene). This was achieved by visual examination of each predicted hotspot (IS positions and orientation) in IGV combined with the corresponding RNA-seq alignments (RNA-seq data of ovine primary B cells). Predicted hotspots that overlapped multiple genes or novel unannotated transcripts were discarded for further analysis in this study. Of the genes uncovered in the genic hotspot class after curation, 618 showed evidence of 1 hotspot while 56 carried multiple hotspots (2–4). Multiple hotspots in the same gene were merged. This resulted in a curated list of 674 genic and 48 intergenic hotspots. Besides genic and intergenic types, we identified hotspots that combined genic and upstream intergenic integrations for the same gene. For each given hotspot a proviral orientation ratio was calculated by counting the number of proviruses integrated in the same orientation relative to the total number of IS within that hotspot, considering the predominant orientation. Hotspots significantly biased towards non-random orientation were defined as these associated with an FDR < 0.1 (one-tailed binomial test). The provirus integration orientation relative to host gene transcription (concordant/discordant) was determined for hotspots that showed a significant orientation bias. To measure the non-randomness of provirus orientation, we also determined the relative orientation and distance of each possible IS pair across all (i) genic and (ii) intergenic IS identified, grouped the results by distance in 100 bp increment bins and computed the orientation ratio (same/opposite) relative to each bin.

To verify that genic hotspots were independent of gene size and gene expression level, the number of IS per gene in our data set (42,113 genic IS, corrected total number of unique genic integrations obtained after manual curation of hotspots defined by simulation) was counted (IS.real(i) = number of IS in gene i) and a simulation (N = 100,000 iterations) was performed as follows: for each iteration IS were assigned to a subset of 10,679 genes matched for expression level with the 674 genic hotspot genes (considering bins of 100 genes centred on each hotspot gene in terms of expression level defined by the average TPM across all ovine samples; ENSEMBL OAR3.1 v84: 25,197 genes), each IS having a probability to be assigned to a gene proportional to both gene size and gene expression level in ovine lymphocytes. The number of simulated IS per gene was counted at each iteration (i.e., IS.sim(i,j) = number of simulated IS in each iteration j), and a P-value that reflects the independence of the number of IS and both gene size and gene expression level was computed for each gene and N = 100,000 iterations comparing IS.real to the distribution of simulated IS.sim, i.e., for gene i : $p(i) = (\# \, IS.sim(i,j) \geq IS.real(i))/100,000$. The proportion of genic hotspots not explainable by gene size and expression level alone ($\pi_1$1) was estimated on the set of nominal P-values obtained for the 674 genic hotspots[64].

**Testing enrichment of abundant infected clones in hotspots.** To assess the putative enrichment of abundant clones in hotspots of integration, we separated the total IS set identified in asymptomatic BLV-infected sheep (66,557 IS) in two classes (abundant IS: Ab + and non-abundant IS: Ab-) according to the number of sequencing reads supporting each integration site. Increasing thresholds corresponding to sequencing read numbers per IS were used (six groups, from minimum 2 reads to minimum 100 reads) to assign an IS to the abundant or non-abundant class of clones. The number of Ab + and Ab- clones located within or outside the 722 defined hotspots respectively was reported for each test group and a one-tailed Fisher's exact test was performed to assess the statistical enrichment of abundant clones in hotspots of integration.

**Statistical assessment of host gene recurrence.** The level of gene recurrence in the vicinity of provirus integration sites across tumours was assessed as follows: protein-coding genes in a 1 Mb genomic window upstream and downstream of each distinct proviral integration site (92 IS) were identified using Bedtools intersectBed tool[62] with ENSEMBL v84 annotation. We calculated unweighted and weighted global recurrence scores by summing individual gene recurrence scores (unweighted score = 1 regardless of the number of occurrences across the 92 IS window-based gene lists; weighted score = number of occurrences of a particular gene across 92 IS window-based gene lists). Observed scores were tested against N = 100,000 simulated recurrence scores obtained from 92 random sets of adjacent genes of same size distribution. We assessed recurrence by P-value counting the frequency of simulated recurrence scores equal to or higher than the observed tumour provirus window-specific recurrence scores, divided by the number of simulated gene lists (N = 100,000). Score calculation and simulations were conducted using R 3.1.1.

Gene recurrence between viral transcript-interacting host genes identified in asymptomatic sheep and tumour samples was assessed as follows: we calculated a gene recurrence score by counting the number of overlapping genes between the 82-gene tumour gene list and the 723-gene asymptomatic gene set and tested the statistical significance of the overlap by simulation based on N = 100,000 recurrence scores obtained from 82 and 723 random or expression level matched simulated gene lists, respectively. Simulated expression-matched gene lists were generated according to expression bins that each consisted of a group of 500 genes

that most closely matched the listed genes expression level based on average TPM computed using RSEM[65] across all available human leukaemia samples. Additional scores were computed based on the recurrence of paralog genes. We assessed recurrence by P-value counting the frequency of simulated recurrence scores equal to or higher than the observed tumour/asymptomatic recurrence scores, divided by the number of simulated gene lists (N = 100,000). Gene recurrence between genic hotspots of proviral integration in asymptomatic sheep samples and provirus-interacting host genes identified in either asymptomatic or tumour samples, as well as HTLV-1 genic integrations retrieved from the public RID database[50], was assessed based on the same simulation method. The level of recurrence between asymptomatic individual samples was computed based on the same method testing unweighted and weighted scores (as defined above) to N = 100,000 size-matched random or expression level-matched simulated gene list scores.

**Cancer driver gene enrichment analysis.** Gene sets were tested for cancer-driver enrichment by calculating a cancer driver enrichment score (ES) using seven publicly available cancer-driver gene lists (CGL, Supplementary Table 2 and Supplementary Fig. 2). Each gene in a CGL is assigned a score that rates its cancer-driver potential. Observed scores were compared to simulated scores obtained from N = 100,000 size-matched random or expression-matched gene sets, including information about paralogs (Random Para, Expr Para) or not (Random, Expr). Simulated expression-matched gene lists were generated as described for assessment of gene recurrence. Direct and paralog ES were obtained by summing each CGL-associated gene score (direct-score), or associated gene's paralog score (paralog-score), respectively. Paralog scores were calculated by multiplying the direct score of the paralog gene with the percentage amino acid identity gene/paralog gene using the Paralog Mapping Table (ENSEMBL). We assessed enrichment by P-value by counting the frequency of simulated ES equal to or higher than the observed gene ES divided by the number of simulation iterations (N = 100,000). In addition, we calculated a global enrichment score ('meta-analysis') that incorporated all seven CGL scores by summing the seven CGL-associated P-values and comparing this meta-P-value to N = 100,000 simulated gene list meta-P-values. This resulted in five empirical 'global P-values'. Score and P-value calculations, as well as simulations, were conducted using R 3.1.1.

**RNA-seq based gene expression analysis.** Gene and exon-specific read counts were processed according to a two-pass normalization step. First read counts were normalized to sequencing depth using the R package-implemented DESeq2. Normalized read counts of the fusion-affected host gene (or specific exons) in a particular tumour sample were divided by the mean normalized read count of this particular gene (or corresponding exons) in all remaining samples of the same host that do not have proviral integration in that locus. This resulted in an average expression value of 1 for the control tumour samples and a normalized expression value for the tumour containing the fusion-affected gene. Statistical significance of expression levels was assessed using Mann–Whitney U-tests.

**Validation of virus–host chimeric transcripts by RT–PCR.** RNA was treated with TURBO DNA-free Kit and retrotranscribed using SuperScript First-Strand Synthesis System for RT–PCR (ThermoScientific) and random hexamers according to the manufacturer's instructions. BLV-host chimeric transcripts were amplified from cDNA using combinations of a forward primer in BLV AS exon 1 and reverse primers binding downstream of the fusion breakpoint identified by RNA-seq (primers from Integrated DNA Technologies; Supplementary Data 5) and generated using Primer 3 (http://bioinfo.ut.ee/primer3/). Two microlitres cDNA was mixed with the PCR mix (4,125 μl $H_2O$, 1,25 μl of each 2.5 nM reverse and forward primer, 0.2 μl 10 mM dNTP (Promega), 2 μl 5 × Q5Reaction Buffer (New England BioLabs) and 0.1 μl Q5High-Fidelity DNA Polymerase (New England BioLabs)). PCR consisted of 35 cycles of 8 s at 98 °C, 20 s at 67 °C and 35 s at 72 °C. Products were visualized on a 2% agarose gel and sequenced by conventional Sanger methods.

**Proviral load quantification.** DNA was isolated using the Qiagen AllPrep DNA/RNA/miRNA kit and proviral DNA was quantified by real-time PCR using primers targeting either the BLV or HTLV-1 3′ region and RPS9 or Actin, respectively, for normalization (primers from Integrated DNA Technologies; Supplementary Data 5). Runs were performed in a 50 μl volume containing 1 μg of total DNA, primers and probe (200 nM concentration of each) in 1 × PCR buffer (Platinum Quantitative PCR SuperMix-UDG) (HTLV-1) or 10 μl containing 50 ng DNA and 1 × Universal PCR Master Mix, No AmpErase UNGa (ThermoScientific) (BLV). Thermocycling conditions were 10 min at 95 °C, followed by 50 cycles at 95 °C for 15 s and 60 °C for 1 min. Standard curves were generated using serial dilutions of DNA from the YR2 cell line (BLV, two proviral copies) or the Tarl2 cell line (HTLV-1, single proviral copy). Proviral load in % PBMCs = (Sample Average Quantity) × 2/(Sample RPS9 or Actin quantity) × 100. The YR2 chromosome that carries the BLV provirus integration appears to be duplicated.

**Testing statistical power of hotspot and cancer driver analysis.** Assuming 20,000 genes of which $y$ are cancer drivers (500–3,000) and $z$ is their fraction present in cancer driver lists (25–100%), we first randomly assigned $x$ integration sites (IS, from 1,000 to 100,000) to $G = 20,000$ genes and reported the maximum number of IS across all genes (max_hits_R) and the sum of IS across all cancer-driver genes (cancerDriverHits_R). We performed $N = 100,000$ iterations resulting in two $N = 100,000$-element vectors representing non-preferential integration (hotspot detection = max_hits_R and cancer-driver enrichment = cancerDriverHits_R). We then assumed the presence of two types of IS, $w$ corresponding to expanded clones (from 0.01 to 100% of the $x$ IS, integrated in cancer drivers) and $1 - w$ to infected but non-expanded clones. We randomly assigned $w$ IS to a fraction $z$ of $y$ cancer-driver genes and $1 - w$ IS to $G = 20,000$ genes and reported (i) the sum of IS across the $y$ cancer-driver genes (cancerDriverHits_Exp) and (ii) the maximum number of IS across all $G = 20,000$ genes (max_hits_Exp) for $N = 100,000$.

$P$-values of max_hits_Exp and cancerDriverHits_Exp were calculated for all parameter combinations:
$$\text{P-value}_{hotspot} = \frac{\sum_{i=1}^{N=100,000} (\text{max\_hits\_Exp} \leq \textbf{max\_hits\_R})}{N = 100,000}$$
and
$$\text{P-value}_{CancerDriverEnrichment} = \frac{\sum_{i=1}^{N=100,000} (\text{cancerDriverHits\_Exp} \leq \textbf{cancerDriverHits\_R})}{N = 100,000},$$
respectively.

We performed $T = 10,000$ iterations and computed the power of hotspot detection and cancer-driver enrichment as $power_{hotspot} = \frac{\sum_{i=1}^{T=10,000} (\text{P-value}_{hotspot} \leq P_{limit})}{T = 10,000}$ and $power_{cancerDriverEnrichment} = \frac{\sum_{i=1}^{T=10,000} (\text{P-value}_{CancerDriverEnrichment} \leq P_{limit})}{T = 10,000}$ for $P_{limit} = 0.01$.

**Statistical analyses.** Analyses of significance for RNA-seq-based gene expression were performed using two-sided Mann–Whitney $U$-tests implemented in R 3.1.1, assuming equal variances. One-tailed Fisher's exact-tests were used to assess recurrence between gene groups and the statistical enrichment of abundant clones in hotspots of integration. Values of $P < 0.05$ were considered as statistically significant. Continuous biological variables were assumed not to follow a normal distribution.

**Data availability.** Sequence data that support the findings of this study have been deposited in the European Nucleotide Archive (ENA) hosted by the European Bioinformatics Institute (EMBL-EBI) and are accessible through study accession number PRJEB19394. All other relevant data are available within the article and its Supplementary Information files or from the corresponding author upon reasonable request.

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

## Acknowledgements

This work was supported by the Fonds de la Recherche Scientifique (FRS), Télévie, les Amis de l'Institut Bordet, the International Brachet Stiftung (IBS), the Fondation Lambeau-Marteaux, the Ligue contre le cancer, l'Institut National du Cancer (INCA), the Cancérople d'île de France and a Télévie Grant to V.H. M.A. holds a Postdoctoral Researcher fellowship of the FRS, A.M. is recipient of a doctoral fellowship from the Institut National du Cancer (INCA), N.R. and K.D. are Scientific Research Worker of Télévie. We thank Wouter Coppieters, Latifa Karim and the GIGA Genomics Platform for sequencing services and support, Franck Mortreux (ENS Lyon, France) for providing samples and cell lines used in this work, Charles Bangham (Imperial College, London, UK) for providing the Tarl2 cell line and Dominique Bron (Institut Jules Bordet, ULB) for comments on the manuscript.

## Author contributions

K.D. performed HTS library preparations and sequencing validations, carried out all sample extractions and analysed the data. M.A. carried out HTS-based integration mapping and clonality analyses, V.H. determined proviral loads in animals and RT–PCR validations, N.R. processed the sequencing data and performed bioinformatics analyses, A.M. and O.H. collected clinical data and provided patient materials, V.A. determined patients' proviral loads, P.G. and N.A. collected and provided animal samples, A.B. and C.C. contributed to data analysis and review of the manuscript, N.R., K.D., M.G. and A.V. performed data analyses and generated the text and the figures, A.V. and M.G. designed and supervised the study. All authors contributed to the final manuscript.

## Additional information

**Competing interests:** The authors declare no competing financial interests.

