## [Peer Review File · Nature Communications]

Reviewer comments:

Reviewer #1

This manuscript utilizes RNA-seq analyses to identify that proviral integration of HTLV-1 and BLV retroviruses is non-random and occurs in specific genomic regions which perturb expression of oncogenes and tumor suppressors to promote malignant transformation. The data add further details to the literature regarding the oncogenic mechanisms of these retroviruses. There are several issues to be addressed that would improve the content of this manuscript as follows:

-While the analysis of asymptomatic BLV-infected animals identifying similar proviral integration as the fully transformed tumors is helpful, it would be very helpful to provide some information on what is distinct between these 2 stages of disease as this is not clear from the current manuscript. Are there additional genetic changes present in fully transformed tumors compared to cells in early stage of disease?

-While the concept of HTLV-1 and BLV integrating in non-random regions of the genome to promote cancer is well described here and a helpful point, the concept of viral integration disrupting cis-gene expression to promote cancer is not new. The authors should therefore avoid using the phrase in the abstract "upset the dogma."

-The manuscript is hard to read and there is an excessive listing of sample identifiers and specific numbers throughout. Dividing the manuscript into Introduction, Results, and Discussion sections with section headers for the Results section would certainly help improve readability.

Reviewer #2 (Remarks to the Author)

The manuscript reports the results of analyses of the provirus integration sites of HTLV-1 or BLV and those of RNA seq using human clinical samples and bovine and sheep samples. The major claim of the paper is reappraisal of the promoter insertion model for the leukemogenesis by HTLV-1/BLV. The results suggested a biased and recurrent distribution of integration sites in the vicinity of host cancer driver genes. Furthermore, the results suggest the provirus integration affect these genes by transcript termination or anti-sense transcription dependent perturbation. Collectively, these data provided supportive evidence for necessity to re-consider the old hypothesis from a new viewpoint.

The conclusion is robust, although it is based on observational studies. Validation by biological experiments remains to be conducted, which appears to be beyond the scope of this report. The results are considered novel because they justify re-appraisal of the insertional mutagenesis model in the context of pathogenicity of this retrovirus family. Experiments are well designed and the interpretation of the results appears to be logical and reasonable. The statistic analyses appear to be properly applied. However, readers may have some difficulties to understand the data presentation, which could be improved so that non-specialist can understand more easily. Another important evidence provided by this manuscript is complete absence of the viral sense transcript in the cells analyzed, which is in accordance with the results reported by Kataoka et al. (Nat Genet 2015;47:1304-1315) and unpublished observations of many researchers. This is evidence against the idea that Tax may be functional in the ATL tumor cells.

As for a part of the results presented in this paper, similar findings have previously been reported by Kataoka et al. (Nat Genet 2015;47:1304-1315) and Satou et al (PNAS 2016;113:3054-3059). However, the authors of latter papers did not go into the claims or discussion about reappraisal of the promoter insertion model of HTLV-1/BLV leukemogenesis.

There may be another way for description of the anti-sense transcripts with various structures not using the name of HBZ or AS, since some of the transcripts only contain exon1 of the HBZ or AS. This means that these are the transcripts driven by the 3LTR anti-sense promoter that is not necessarily described as variants of HBZ or AS.

I have checked reviewers' comments and responses by the authors. In conclusion, I think that the authors responded faithfully and appropriately to the comments by the reviewers in a convincing way. Regarding the minor specific comments, the authors also appear to respond appropriately to the reviewers' comments.

I have an impression that some parts of the reviewers' comments are influenced by their misunderstanding and/or preconception.

I think that authors have learned the previously published results in the related fields and correctly understood their significance and limitations. Therefore, I do not find any problem in authors' responses to the reviewers.

As for the apparent discrepancy in the trends of integration sites between BLV and HTLV-1, I speculate another possibility that may have made differences. That is the timing of infection and initial proliferation of infected cells. ATL is generally believed to occur among the carriers of mother-to-child transmission, which means that the infected cell population or clones may have established when the host immunity is immature. This idea is not based on the direct evidence, thus, it remains to be tested by experiments in the future

Toshiki Watanabe, M.D., Ph.D.

Point-by-point response to issues raised by the referees

Reviewer #1 is one of the original reviewers from *Nature Genetics*

Reviewer #2 was recruited to comment on the remarks of the original reviewers #2 and #3 who did not comment in the second round of review.

Reviewer #1

This manuscript utilizes RNA-seq analyses to identify that proviral integration of HTLV-1 and BLV retroviruses is non-random and occurs in specific genomic regions which perturb expression of oncogenes and tumor suppressors to promote malignant transformation. The data add further details to the literature regarding the oncogenic mechanisms of these retroviruses. There are several issues to be addressed that would improve the content of this manuscript as follows:

-While the analysis of asymptomatic BLV-infected animals identifying similar proviral integration as the fully transformed tumors is helpful, it would be very helpful to provide some information on what is distinct between these 2 stages of disease as this is not clear from the current manuscript. Are there additional genetic changes present in fully transformed tumors compared to cells in early stage of disease?

Differences between the asymptomatic stage and the fully transformed tumors:

BLV infects B-cells, and following a transient phase of horizontal replicative dissemination, primarily spreads via **polyclonal** expansion, producing many long lived clones. As a result, the chronic **asymptomatic** stage of infection is characterized by a **large number of clones of varying abundance**, each uniquely identified by their proviral integration site in the genome. Following a protracted incubation period, for unknown reasons one of these clones rapidly expands, leading to the aggressive monoclonal leukemia/lymphoma. While multiple integration sites characterize the chronic stage of infection, acute tumors are characterized by the presence of a single **dominant clone**, with an underlying polyclonal population of infected cells. In the majority of the tumors (~ 87 %, see our manuscript, line 10 in the *Results* section), this clone shows a **single proviral integration**.

In summary, the clones examined by HTS mapping of proviral integration sites in asymptomatic individuals (*Results*, identification of hotspots) can be defined as “non-malignant” clones of low abundance that have not yet acquired the “events” (or sufficient events) that will ultimately cause the switch to full-blown malignancy.

In the manuscript:

We have provided a better description of the asymptomatic and tumor stages in the *Introduction* (new section in the revised manuscript): “*In chronic stages of infection, HTLV-1 and BLV propagate primarily through clonal expansion of infected T- or B-cells respectively, resulting in the presence of multiple clones of varying abundance each uniquely identified by their proviral integration site in the host genome. Following a protracted incubation period, one of these clones expands, leading to the accumulation of malignant cells in the peripheral blood (leukemia) and/or diverse tissues (lymphoma)^{4,6-8}. Tumor cells consist of a predominant malignant T- or B-cell clone and chiefly harbour a single integrated provirus, yet integration sites are very variable⁸⁻¹⁰.*”

The “asymptomatic” samples from BLV infected sheep are described in *Methods* (“samples”) and in Supplementary Data 1 (ovine samples). In the *Results* section, we have mentioned the polyclonal non-malignant nature of the samples collected at early asymptomatic stages: “*We first comprehensively analyzed proviral integration sites at early non-malignant stages (characterized by the presence of multiple clones of low abundance). This was achieved by very deep, high-throughput DNA sequencing based mapping of BLV integration sites for 10 infected but still asymptomatic sheep (proviral load range: 0.02-34 %, clone abundance range: 0.002-9.524 %, Supplementary Data 1).*”

Finally, we have provided information with regards to genetic changes in fully-transformed BLV tumors in the *Introduction*:

“In the fraction of infected individuals that do progress, many years separate the initial infection from the development of leukemia/lymphoma. This indicates that infection with BLV/HTLV-1 is not sufficient to provoke tumor development and that secondary events are required to make the transition to a neoplasm. A recent study examined the landscape of mutations in ATMs and found frequent alterations enriched in T cell–related pathways and immunosurveillance⁹. As regards BLV-induced tumors, beyond limited studies that reported frequent genome instability and mutation of p53^{24,25} the occurrence of secondary events in BLV malignancies remains largely unexplored.”

-While the concept of HTLV-1 and BLV integrating in non-random regions of the genome to promote cancer is well described here and a helpful point, the concept of viral integration disrupting cis-gene expression to promote cancer is not new. The authors should therefore avoid using the phrase in the abstract "upset the dogma."

We have removed this phrase from the abstract.

-The manuscript is hard to read and there is an excessive listing of sample identifiers and specific numbers throughout. Dividing the manuscript into Introduction, Results, and Discussion sections with section headers for the Results section would certainly help improve readability.

The revised manuscript is now divided into *Introduction*, *Results*, and *Discussion* sections with section headers for the *Results* section.

Reviewer #2 (Remarks to the Author):

The manuscript reports the results of analyses of the provirus integration sites of HTLV-1 or BLV and those of RNA seq using human clinical samples and bovine and sheep samples. The major claim of the paper is reappraisal of the promoter insertion model for the leukemogenesis by HTLV-1/BLV. The results suggested a biased and recurrent distribution of integration sites in the vicinity of host cancer driver genes. Furthermore, the results suggest the provirus integration affect these genes by transcript termination or anti-sense transcription dependent perturbation. Collectively, these data provided supportive evidence for necessity to re-consider the old hypothesis from a new viewpoint.

The conclusion is robust, although it is based on observational studies. Validation by biological experiments remains to be conducted, which appears to be beyond the scope of this report. The results are considered novel because they justify re-appraisal of the insertional mutagenesis model in the context of pathogenicity of this retrovirus family. Experiments are well designed and the interpretation of the results appears to be logical and reasonable. The statistic analyses

appear to be properly applied. However, readers may have some difficulties to understand the data presentation, which could be improved so that non-specialist can understand more easily. Another important evidence provided by this manuscript is the complete absence of the viral sense transcript in the cells analyzed, which is in accordance with the results reported by Kataoka et al. (Nat Genet 2015;47:1304-1315) and unpublished observations of many researchers. This is evidence against the idea that Tax may be functional in the ATL tumor cells.

As for a part of the results presented in this paper, similar findings have previously been reported by Kataoka et al. (Nat Genet 2015;47:1304-1315) and Satou et al (PNAS 2016;113:3054-3059). However, the authors of latter papers did not go into the claims or discussion about reappraisal of the promoter insertion model of HTLV-1/BLV leukemogenesis. There may be another way for description of the anti-sense transcripts with various structures not using the name of HBZ or AS, since some of the transcripts only contain exon 1 of the HBZ or AS. This means that these are the transcripts driven by the 3LTR anti-sense promoter that is not necessarily described as variants of HBZ or AS.

We agree with Dr. Watanabe that the virus-host chimeric transcripts, although dependent on the 3'LTR antisense promoter activity, cannot be considered as variants of *HBZ* or *AS1/AS2* transcripts. We have renamed these transcripts throughout the manuscript and have cited them either as “3'LTR dependent chimeric transcripts”, “antisense RNA dependent hybrid transcripts” or “3'AS-dependent virus-host transcripts”. The terms *HBZ* and *AS1/2* are still used when pointing to splicing events that involve *exon 1* or *exon 2* (i.e. “capture of host exons located upstream of the provirus by the first *HBZ/AS* exon” and “capture of viral *HBZ/AS* exon 2 by host gene transcripts”).

I have checked reviewers' comments and responses by the authors. In conclusion, I think that the authors responded faithfully and appropriately to the comments by the reviewers in a convincing way. Regarding the minor specific comments, the authors also appear to respond appropriately to the reviewers' comments.

I have an impression that some parts of the reviewers' comments are influenced by their misunderstanding and/or preconception.

I think that authors have learned the previously published results in the related fields and correctly understood their significance and limitations. Therefore, I do not find any problem in authors' responses to the reviewers.

As for the apparent discrepancy in the trends of integration sites between BLV and HTLV-1, I speculate another possibility that may have made differences. That is the timing of infection and initial proliferation of infected cells. ATL is generally believed to occur among the carriers of mother-to-child transmission, which means that the infected cell population or clones may have established when the host immunity is immature. This idea is not based on the direct evidence, thus, it remains to be tested by experiments in the future.

Toshiki Watanabe, M.D., Ph.D.

We thank Dr. Watanabe for these positive comments. We also thank him for having agreed on commenting on the criticisms of the original reviewers and for having expressed his opinion so clearly.